

# Growing topography due to contrasting rock types in a tectonically dead landscape

Daniel Peifer[1,2], Cristina Persano[1], Martin D. Hurst[1], Paul Bishop[1], Derek Fabel[3]

[1]School of Geographical and Earth Sciences, University of Glasgow, Glasgow, G12 8QQ, UK
5   [2]CAPES Foundation, Ministry of Education of Brazil, Brasilia - DF 70040-020, Brazil
[3]Scottish Universities Environmental Research Centre, East Kilbride, G75 0QF, UK

*Correspondence to*: Daniel Peifer (peiferdaniel@gmail.com)

**Abstract.** Many mountain ranges survive in a phase of erosional decay for millions of years (Myr) following the cessation of tectonic activity. Landscape dynamics in these post-orogenic settings have long puzzled geologists due to the expectation that topographic relief should decline with time. Our understanding of how denudation rates, crustal dynamics, bedrock erodibility, climate, and mantle-driven processes interact to dictate the persistence of relief in the absence of ongoing tectonics is incomplete. Here we explore how lateral variations in rock type, ranging from resistant quartzites to less-resistant schists and phyllites and up to the least-resistant gneisses and granitic rocks, have affected rates and patterns of denudation and topographic forms in a humid semitropical, high-relief, post-orogenic landscape in Brazil where active tectonics ended hundreds of Myr ago. We show that denudation rates are negatively correlated to topographic relief, channel steepness and modern precipitation rates. Denudation instead correlates with inferred bedrock strength, with resistant rocks denuding more slowly relative to more erodible rock units, and suggest that the efficiency of fluvial erosion varies primarily due to these bedrock differences. Variations in erodibility continue to drive contrasts in rates of denudation in a tectonically inactive landscape evolving for hundreds of Myr, suggesting that equilibrium is not a natural attractor state and that relief continues to grow through time. Over the long timescales of post-orogenic development, exposure at the surface of rock types with differential erodibility can become a dominant control on landscape dynamics by producing spatial variations in geomorphic processes and rates, promoting the survival of relief, and determining spatial differences in erosional response timescales long after cessation of mountain building.

## 1 Introduction

25   The question of how landscapes evolve in the aftermath of mountain building has intrigued geomorphologists since the early stages of the discipline, and classic concepts such as the cycle of erosion (Davis, 1899) and dynamic equilibrium landforms (Hack, 1960) were defined in the context of these post-orogenic landscapes (Bishop, 2007). In particular, reasons for the persistence of high topographic relief in ancient mountain belts for many millions of years (Myr) after crustal thickening has ceased remain enigmatic. We know with a reasonable degree of certainty that net erosion in these landscapes and the resulting

30   rebound of the underlying lithosphere by isostasy are central mechanisms controlling the extended longevity of post-orogenic



landforms (Gilchrist and Summerfield, 1990; Bishop and Brown, 1992; Bishop, 2007). However, a range of other factors and interactions play essential roles in the post-orogenic evolution of ancient landscapes, including variations in bedrock incision dynamics (e.g., Baldwin et al., 2003; Egholm et al., 2013), mantle-flow dynamics and its influence on the overlying crust (e.g., Gallen et al., 2013; Liu, 2014), vertical and lateral variations in bedrock erodibility (e.g., Twidale, 1976; Bishop and Goldrick, 35 2010; Gallen, 2018; Bernard et al., 2019; Vasconcelos et al., 2019), densification of the lower crust and resulting reduction of the buoyancy of the lithosphere (e.g., Blackburn et al., 2018), and tectonic uplift in response to far-field stresses (e.g., Hack, 1982; Quigley et al., 2007).

Examples of ancient mountain belts marked by high elevations and steep slopes include the Appalachian Mountains, several mountain ranges SE Brazil, the Cape Mountains, the Lachlan Fold Belt in SE Australia, the Ural Mountains, the Caledonides, 40 the Western Ghats, and the Sri Lanka orogen (Fig. 1). These high-relief post-orogenic settings are associated with different climate conditions, tectonic histories, effective elastic thicknesses of the lithosphere, and geological architectures. For instance, some of these ancient mountain belts are located in a passive margin context, whereas others are located in the deep interior of the continents (Fig. 1). Yet they share common geomorphic characteristics such as overall low rates of denudation (e.g., Harel et al., 2016), relative Cenozoic tectonic quiescence (e.g., Twidale, 1976; Mandal et al., 2015), exposure of a variety of 45 resistant and more erodible lithologies (e.g., Bishop and Goldrick, 2010; Gallen, 2018; Vasconcelos et al., 2019), peak elevations that may exceed 2000 m and average elevations commonly higher than 1000 m (e.g., Gallen et al., 2013; Scharf et al., 2013; von Blanckenburg et al., 2004).

Early landscape evolution schemes (e.g., Davis, 1899) and quantitative estimates of post-orogenic relief reduction (e.g., Ahnert, 1970; Baldwin et al., 2003; Egholm et al., 2013) predict a progressive decay in both topographic relief and denudation 50 rates, with residual post-orogenic landforms marked by featureless topography reminiscent of peneplains after hundreds of Myr of ongoing denudation. More recently, a range of geomorphic and thermochronologic data indicates that topographic evolution in ancient mountain chains may be more dynamic than otherwise expected, with different types of forcing (e.g., tectonic, climatic, lithologic, mantle-driven) affecting at least some of these landscapes in post-orogenic times (e.g., Pazzaglia and Brandon, 1996; Quigley et al., 2007; Gallen et al., 2013; Tucker and van der Beek, 2013; Liu, 2014; Gallen, 2018). Our 55 current knowledge on post-orogenic landscape evolution suffers from an incomplete understanding of how and to what extent different types of forcing may act in concert in driving the development of decaying mountain belts that are evolving over Myr timescales (Bishop, 2007; Tucker and van der Beek, 2013).

Most post-orogenic landscapes are characterised by complex and spatially variable lithology, often including crystalline rocks, different types of deformed metasediments and sedimentary covers, and volcanic units (e.g., Dorr, 1969; Bierman and Caffe, 60 2001; Barreto et al., 2013; Gallen et al., 2013; Mandal et al., 2015). Spatial variations in rock type have long been identified as a critical factor in post-orogenic development for they determine spatial heterogeneities in erodibility (e.g., Hack, 1960; Dorr, 1969; Hack, 1975; Twidale, 1976; Mills, 2003; Bishop and Goldrick, 2010; Gallen, 2018; Bernard et al., 2019; Vasconcelos et al., 2019). In particular, correlations between rock type and topographic forms were observed in post-orogenic



landscapes, with high topographic relief and steep channel reaches associated with resistant rocks (e.g., Hack, 1960, 1975;
Twidale, 1976; Mills, 2003; Spotila et al., 2015; Gallen, 2018), and such links were interpreted in several cases as a dynamic
equilibrium adjustment, likely driven by denudational isostatic rebound, where spatial variations in topographic relief persist
over time as a function of contrasts in bedrock erodibility and denudation rates are spatially uniform (e.g., Hack, 1960; Matmon
et al., 2003; Scharf et al., 2013; Mandal et al., 2015). In contrast, recent modelling studies indicate that exposure at the surface
of rock units with substantial differences in rock strength dictate complex patterns of denudation, with significant spatial and
temporal variations in denudation rates and possibly the persistence of non-steady-state conditions as long as different rock
units are exposed (Forte et al., 2016; Perne et al., 2017). Post-orogenic settings are well-suited as natural laboratories to explore
further the role of spatial variations in lithology in landscape evolution as these are lithologically heterogeneous landscapes
that last experienced major active rock uplift tens to hundreds of millions of years ago (Bishop, 2007). Nevertheless, few
studies have directly explored the spatial variability of denudation rates in post-orogenic settings as a function of the full
spectrum of variations in underlying lithology and topographic relief in these landscapes.

Here, we investigated how denudation rates vary spatially in a humid semitropical, high-relief post-orogenic area in Brazil
where the last phase of tectonic activity ended ~500-450 Myr ago and explored the relationships between denudation rates and
topographic relief, channel steepness, precipitation rates and rock type. Denudation rates were measured using [10]Be
concentrations in fluvial sediments from catchments spanning the range of topographic relief and bedrock lithologies in the
study area.

## 2 Geological setting

The study area is the Quadrilátero Ferrífero (Brazil), one of the highest elevation areas in Brazil, with a peak elevation of 2,076
m. The name Quadrilátero Ferrífero (QF) translates as 'Iron Quadrangle', referring both to its vast iron ore reserves and the
roughly rectangular alignment of its ridges (Dorr, 1969). Local relief can reach 1,189 m over a 2-km diameter circular window
(Fig. 2A). There is an abundance of mixed bedrock-alluvial channels that deeply cut into the most resistant rocks, but are less
incised where more erodible lithologies are exposed; everywhere, however, the slope is enough for the detrital material to be
removed, as alluvium has not significantly accumulated in the study area (Dorr, 1969). Normalised channel steepness, which
is a metric for channel slope normalised to catchment size according to an assumed channel profile concavity, differs over
three orders of magnitude (Fig. 2C). Mean annual precipitation rates vary from 1356 to 1729 mm/yr (Fick and Hijmans, 2017),
and the mean annual temperature is ~20°C (Dorr, 1969).

Resistant and more erodible lithologies are exposed in a complex geological pattern (Fig. 2B) that reflects a polyphase
deformation history, the last episode of which was ~500-450 Myr ago (Dorr, 1969; Chemale Jr et al., 1994; Alkmim and
Marshak, 1998). The exposed lithologies comprise principally Archean and Paleoproterozoic sequences, usually
metamorphosed and steeply dipping (≥35°), including gneisses and granitic rocks, schists, phyllites, quartzites,



metaconglomerates, metacarbonate rocks, metavolcanics, banded iron formations, and iron duricrusts (Dorr, 1969; Chemale Jr et al., 1994; Alkmim and Marshak, 1998). An array of geochronological data imply that the current topography is long-lived, these include: goethite (U-Th)/He ages and cosmogenic $^3$He concentrations indicating laterite development since 55 Ma (Monteiro et al., 2018); ~70 Ma $^{40}$Ar/$^{39}$Ar ages in weathering profiles (Vasconcelos and Carmo, 2018); and low denudation rates (<3 m/Myr) inferred by cosmogenic $^3$He and $^{10}$Be inventories (Monteiro et al., 2018; Salgado et al., 2008). However,
some authors hypothesised that the eastern part of the QF was affected by Cenozoic tectonics based principally on field evidence of post-depositional deformation in Cenozoic units that fill small basins (e.g., Sant' anna et al., 1997).

## 3 Methods

### 3.1 Determination of denudation rates

We collected alluvial sand from the bed of 25 active channels from catchments spanning the range of topographic relief in the
QF (Fig. 2A) and the range of bedrock lithologies (Fig. 2B), from the resistant quartzites to the least-resistant (under humid semitropical conditions) gneisses and granitic rocks, for the determination of denudation rates from detrital $^{10}$Be concentrations. The sampled basins do not show evidence of deep-seated landslides, and there are no records of significant landslide activity in the study area (Dorr, 1969). Samples were prepared and analysed at SUERC (Scottish Universities Environmental Research Centre), Scotland, following standard procedures (Kohl and Nishiizumi, 1992); NIST SRM4325 was
the $^{10}$Be standard. The resulting $^{10}$Be/$^9$Be ratios for each sample were corrected for processed blank ratios ($n = 2$), ranging between 0.2 and 3.2% of the sample $^{10}$Be/$^9$Be ratios, with uncertainties propagated in quadrature (Balco et al., 2008).

Catchment-averaged denudation rates were derived from $^{10}$Be concentrations using the CRONUS online calculator v. 3.0 (Balco et al., 2008). We used the CAIRN method (Mudd et al., 2016) to quantify catchment-averaged pressure employing a pixel-by-pixel approach and recommended parameters (Mudd et al., 2016). We report denudation rates using the time-
independent Lal/Stone scaling method (Lal, 1991; Stone, 2000), and assuming a standard bedrock density of 2.7 g/cm$^3$ (e.g., von Blanckenburg et al., 2004; Mandal et al., 2015). We have not applied corrections for topographic shielding (DiBiase, 2018). We have also incorporated eight detrital $^{10}$Be-derived concentration measurements previously published for the QF (Salgado et al., 2008) in our dataset; however, we recalculated denudation rates using the same method described above to ensure consistency (Table S1).

### 3.2 Quantification of catchment-averaged geomorphic metrics

We used a 12 m TanDEM-X (TerraSAR-X add-on for Digital Elevation Measurement) DEM to extract the topographic metrics (i) local relief, and (ii) normalised channel steepness. The WordClim v.2 dataset (Fick and Hijmans, 2017) was used to extract (iii) mean annual precipitation rates. These metrics have been demonstrated to play an essential role in revealing the pattern and style of landscape evolution in erosive settings (e.g., Ahnert, 1970; Montgomery and Brandon, 2002; DiBiase et al., 2010;



Portenga and Bierman, 2011; Harel et al., 2016). Catchments were extracted from the DEM following standard hydrological routing procedures using sample locations as pour points. Catchment-averaged values for each metric were determined as the average of all local values within each catchment.

Various authors observed empirically that channel slope ($S$) declines progressively as contributing drainage area ($A$) increases in the downstream direction of a channel profile, in a relationship that can be described by a power-law (e.g., Flint, 1974):

$$S = k_s A^{-\theta}, \tag{1}$$

where $\theta$, termed the 'concavity index', controls how rapidly channel slope decreases with increasing drainage area, and $k_s$, referred to as the 'steepness index', is a measure of channel steepness normalised by upstream drainage area which has been demonstrated to correlate positively with denudation rates in different geomorphic conditions (e.g., DiBiase et al., 2010; Mandal et al., 2015; Harel et al., 2016). The parameters $k_s$ and $\theta$ covary, and this autocorrelation is corrected by defining a

fixed reference concavity index ($\theta_{ref}$) which is used to extract a normalised steepness index ($k_{sn}$) from the data (Kirby and Whipple, 2012).

The Eq. (1) can be used to derive estimates of $k_s$ and $\theta$ (or $k_{sn}$) from regressions of log-transformed slope-area data (Kirby and Whipple, 2012). Alternatively, one can extract $k_{sn}$ using an approach referred to as the integral method (Perron and Royden, 2013). For that, the Eq. (1) is rearranged replacing $S$ for $dz/dx$ and separating these variables, where $z$ is elevation and $x$ is

distance along the channel, which is then integrated in the upstream direction from an arbitrarily base level at $x_b$, resulting in an equation for the elevation profile (e.g., Mudd et al., 2018):

$$z(x) = z(x_b) + \left(\frac{k_s}{A_0^{\theta}}\right) \int_{x_b}^{x} \left(\frac{A_0}{A(x)}\right)^{\theta} dx, \tag{2}$$

where $A_0$ is a reference drainage area that is inserted to make the area term dimensionless (Perron and Royden, 2013). One can then define a longitudinal coordinate chi ($\chi$) with dimensions of length using (Perron and Royden, 2013; Mudd et al., 2014):

$$\chi = \int_{x_b}^{x} \left(\frac{A_0}{A(x)}\right)^{\theta} dx. \tag{3}$$

In the case of $A_0 = 1$, the slope of a longitudinal profile in a $z$ versus $\chi$ plot is the channel steepness index ($k_s$), or the normalised steepness index ($k_{sn}$) if the extraction is based on $\theta_{ref}$. We quantified $k_{sn}$ as the derivative of $\chi$ and $z$ (e.g., Mudd et al., 2014) instead of using regressions of slope-area data (e.g., DiBiase et al., 2010; Kirby and Whipple, 2012) because the integral method does not require estimating slope from the DEM, resulting in more precise $k_{sn}$ values (Perron and Royden, 2013). We

estimated how $\theta$ varies over the study area based on the disorder method (Hergarten et al., 2016), which was carried out using routines developed by Mudd et al. (2018), to derive an optimal $\theta$ value for the entire landscape. The drainage network was extracted using an area threshold of $0.5$ km$^2$ (e.g., Beeson et al., 2017; Campforts et al., 2020), which is lower than the minimum drainage area among the catchments where we collected fluvial sediments ($0.86$ km$^2$), and is a reasonable critical threshold for





the study area (Fig. S1). We computed $\theta$ for all catchments of stream-order higher than third-order following Strahler (1957)
($n = 144$), covering the entire study area, that were extracted using code developed by Clubb et al. (2019). The mean $\theta$ for the QF (0.44; Table 1) is close to the frequently used value of 0.45 (e.g., Mandal et al., 2015), thus we set 0.45 as the reference concavity from which we computed the normalised channel steepness employing the segmentation method of Mudd et al. (2014).

We quantified local relief as the elevation range within a neighbourhood defined by a 2-km diameter circular window. The
choice of the local relief window was based on sensitivity analysis (DiBiase et al., 2010). We compared the goodness-of-fit in bivariate regressions between average values of local relief (with window diameter varying from 0.5 to 5.0 km) and normalised channel steepness for every catchment in the study area with stream-order higher than second-order. In this situation, the local relief obtained using the 2-km diameter window exhibits the highest goodness-of-fit (measured using the ordinary least squares method; Myers, 1990) with catchment-averaged normalised channel steepness (Fig. S2) and, therefore, it is the one that has
been used throughout. Finally, we extracted mean annual precipitation rates from the 30 arc-second spatial resolution WorldClim v.2 global monthly precipitation dataset for the years 1970 to 2000, which is based on raw weather station data and covariates from satellite sensors that were interpolated and combined, creating global climate surfaces (Fick and Hijmans, 2017).

### 3.3 Lithological strength and fluvial erosion efficiency in the QF

The QF is characterised by the presence of vast ore deposits, particularly gold, iron, and manganese (Dorr, 1969; Lobato et al., 2001). The exploitation of these ore reserves has led to focused research, and the QF is the most systematically investigated geological domain of Brazil (Lobato et al., 2001). We extracted geological data from the 'Projeto Geologia do Quadrilátero Ferrífero' dataset mapped at a scale of 1:25,000 (Lobato et al., 2005).

Rivers in the study area erode through a landscape marked by variations in rock type, including granitic, argillaceous,
quartzose, and carbonate rocks, as well as iron formations. These rock units are metamorphosed and steeply dipping as a result of QF's polyphase deformation history (Dorr, 1969; Chemale Jr et al., 1994; Alkmim and Marshak, 1998). There is a clear consensus from geological research that the exposed lithologies have differential resistance to weathering and erosion, whereby quartzites, banded iron formations, and iron duricrusts comprise the most resistant lithologies; schists, phyllites, dolomitic units, and metavolcanics rocks are less-resistant lithologies; and gneisses and granitic rocks are the least-resistant rocks
exposed in the QF (e.g., Dorr, 1969; Salgado et al., 2008; Monteiro et al., 2018; Vasconcelos and Carmo, 2018). Following the approach taken by previous studies (e.g., Lague et al., 2000; Korup, 2008; Jansen et al., 2010; Hurst et al., 2013), we assumed that such classification of rock strength as a function of rock type is valid although we do not have rock strength measurements to support it. Given the humid semitropical condition of the QF, we expect differential resistance to reflect variations in the susceptibility to chemical weathering of the different rock units (Meybeck, 1987; White and Blum, 1995). For
instance, gneisses and granitic rocks with an abundant presence of feldspars are readily weathered whereas physically robust





and chemically inert quartzites weather much slower, via the intergranular solution of quartz (Dorr, 1969). In general, in the study area, quartzites and banded iron formations form pronounced ridges and steep landforms with relatively unweathered outcrops compared to areas in gneisses and granitic rocks associated with broad lowlands that are deeply weathered, with regoliths extending to depths of 50 m or more (e.g. Dorr et al., 1969; Salgado et al., 2008)

Theory and field observations demonstrate that rock strength influences long-term fluvial incision rates in erosive settings (e.g., Gilbert, 1877; Howard and Kerby, 1983; Stock and Montgomery, 1999; Whipple and Tucker, 1999; Jansen et al., 2010; Bursztyn et al., 2015) in concert with a number of other controls, including climate conditions and runoff efficiency, channel width scaling, extreme hydrologic events, and frequency of debris-flow (e.g., Snyder et al., 2000; Whipple et al., 2000a; Kirby and Whipple, 2001; Duvall et al., 2004; Zondervan et al., 2019). All these factors are encapsulated in a dimensional coefficient

of erosion efficiency ($K$) in the commonly used stream-power model (Howard and Kerby, 1983):

$$E = KA^m S^n,  \tag{4}$$

where $E$ is the long-term fluvial erosion, $A$ is the upstream contributing drainage area, $S$ is the local channel gradient, and $m$ and $n$ are positive exponents that depend on incision process, channel hydraulics, and rainfall variability (Whipple and Tucker, 1999).

Whereas most variables in the stream-power model can be derived from DEM data, the fluvial erosion efficiency coefficient ($K$) cannot be measured directly, and thus computing $K$ demands constraints on timing/rates of river evolution (Zondervan et al., 2019). We have a limited understanding of how $K$ varies in different geomorphic conditions and what controls its variability due to the few studies that derived absolute constraints on $K$ and confounding between the multiple controls encoded in $K$ (Snyder et al., 2000; Whipple et al., 2000a; Duvall et al., 2004; Harel et al., 2016; Zondervan et al., 2019). More sophisticated

versions of the stream-power model may explicitly account for different controls in $K$ (e.g., DiBiase and Whipple, 2011; Zondervan et al., 2019; Campforts et al., 2020), yet such approach requires specific data on an adequate resolution such as river hydraulic geometry, rock strength measurements, and hydrological data to resolve spatial and temporal runoff variability, which are not readily available.

We computed catchment-averaged values of $K$ for the QF using an approach similar to Gallen (2018) based on $k_{sn}$ and average

erosion rates. Such an approach requires that spatial variability in long-term rock uplift over the study area is low, which is a reasonable assumption for the QF. Rearranging Eq. (4) to solve for channel slope can cast the stream-power model in the same form as Eq. (1):

$$S = \left(\frac{E}{K}\right)^{1/n} A^{-m/n},  \tag{5a}$$

where:

$$k_{sn} = \left(\frac{E}{K}\right)^{1/n},  \tag{5b}$$





$$\theta_{ref} = m/n, \tag{5c}$$

and thus:

$$E = K\, k_{sn}{}^{n}. \tag{5d}$$

We extracted $k_{sn}$ using 0.45 as the reference concavity and hence derived $K$ values have units of $m^{0.1}$/yr. We assumed that
catchment-wide denudation rates derived from detrital $^{10}$Be concentrations are representative for long-term fluvial incision in
the QF, considering that there are no records of occurrence of deep-seated landslides (Dorr, 1969) and sampled catchments are
of a reasonable size. We calculated $K$ assuming $n = 1$ and $n = 2$, which have been previously demonstrated as feasible values
of $n$ (Lague, 2014). In the study area, the range in precipitation rates is relatively low (from 1356 to 1729 mm/yr), with areas
in higher elevations principally underlain by resistant rocks receiving more precipitation than valley bottoms in more erodible
rock units (Fig. S3), indicating opposing effects in $K$ between inferred rock strength and precipitation rates. We explored how
rock type and precipitation rates are linked to catchment-averaged values of $K$.

## 4 Results

### 4.1 Catchment-averaged denudation rates in the QF

Catchment-averaged denudation rates in the study area range from $0.6 \pm 0.1$ m/Myr to $22.2 \pm 1.9$ m/Myr, with a regional mean
denudation rate of 6.4 m/Myr (Fig. 3; Table S1). Denudation rates in the QF are thus comparable to or lower than previous
estimates of denudation rates in other tectonically inactive, post-orogenic settings (e.g., Harel et al., 2016), including the Cape
Mountains (e.g., Scharf et al., 2013), the Appalachian Mountains (e.g., Matmon et al., 2003), the Ozark dome (e.g., Beeson et
al., 2017), the Sri Lankan uplands (e.g., von Blanckenburg et al., 2004), and the Western Ghats (e.g., Mandal et al., 2015).
However, denudation rates are not uniformly low, but vary by more than one order of magnitude in the study area (Fig. 3),
from the slowly denuding catchments in the eastern part of the QF, where all catchments exhibit denudation rates lower than
3.5 m/Myr, to the western part of the QF where catchments denude at higher rates up to $22.2 \pm 1.9$ m/Myr (Fig. 3).

### 4.2 Links between topographic metrics, precipitation rates, rock type, and denudation rates

We find that denudation rates are negatively correlated with catchment-averaged relief (Fig. 4A), contrary to the established
understanding of links between denudation and topographic relief, and the bulk of supporting empirical studies (e.g., Ahnert,
1970; Montgomery and Brandon, 2002; DiBiase et al., 2010; Harel et al., 2016). Similarly, denudation rates are negatively
correlated with normalised channel steepness (Fig. 4B), which is often used to infer denudation rates based on empirical
evidence for a positive correlation between denudation and normalised channel steepness in tectonically active landscapes
(e.g., DiBiase et al., 2010; Kirby and Whipple, 2012; Harel et al., 2016). We also find a negative relationship between





denudation rates and mean annual precipitation (Fig. 4C), contrary to the expectation that wetter climates drive more rapid
denudation (e.g., Moon et al., 2011; Harel et al., 2016) and, finally, a weak correlation with catchment area (Fig. S4).

Denudation rates can instead be linked to inferred rock strength (Fig. 4E). Catchments underlain by quartzites are associated with the slowest denudation rates (from $0.6 \pm 0.1$ to $3.3 \pm 0.3$ m/Myr, with a mean of 2.2 m/Myr). Catchments in mixed lithologies where ≥60% of catchment area consists of resistant lithologies are denuding at slightly higher rates up to $3.5 \pm 0.3$ m/Myr (with a mean of 2.6 m/Myr). In contrast, catchments in less-resistant schists or mixed lithologies with <60% of resistant
lithologies denude more rapidly. Finally, the low-relief catchments underlain by the least-resistant (under humid semitropical climate conditions) gneisses and granitic rocks, as well as catchments dominated by phyllites, denude at faster rates of up to $22.2 \pm 1.9$ m/Myr. Similarly, we observe that a higher percentage contribution of resistant rocks within the catchment area (i.e., areas in quartzites and banded iron formations) determines lower rates of denudation (Fig. 4D).

The regional distribution of topographic metrics indicates a positive correlation between topography and inferred rock strength
(Fig. 5). In this situation, the high-end of the distribution of topographic metrics, as well as mean and median values, exhibit similar trends of high values for areas underlain by quartzites and banded iron formations, intermediate values for less-resistant rock types such as metabasalts, schists, and phyllites, and low values for areas underlain by the least-resistant basement rocks which are also marked by lower variability in topography than areas dominated by resistant rocks. However, the low-end of the distribution of topographic metrics show comparable low values for every rock type, which indicates that areas marked by
subdued local relief and channel steepness are ubiquitous to all lithologies. We also find positive relationships between catchment-averaged topographic relief and inferred rock strength (Fig. 5C). In particular, we observe that catchments in mixed lithologies where ≥60% of catchment area consists of resistant lithologies exhibit substantially higher local relief than catchments in mixed lithologies with less than 60% of resistant lithologies.

### 4.3 Fluvial erosion efficiency and its relationships with rock type and precipitation rates

Assuming the slope exponent $n = 1$, we find that the fluvial erosion efficiency coefficient ($K$) differs by three orders of magnitude in the study area, varying from $5.8 \times 10^{-9}$ to $1.7 \times 10^{-6}$ m$^{0.1}$/yr, with a global mean of $3.3 \times 10^{-7}$ m$^{0.1}$/yr and a standard deviation of $4.5 \times 10^{-7}$ m$^{0.1}$/yr (Fig. 6; Table S1). Our results indicate that $K$ decreases substantially with increasing inferred rock strength, varying from low $K$ values in areas underlain by quartzites (with a mean $K$ of $3.6 \times 10^{-8}$ m$^{0.1}$/yr) to considerably higher $K$ values in areas in gneisses and granitic rocks (with a mean $K$ of $1.2 \times 10^{-6}$ m$^{0.1}$/yr) (Fig. 6). We observe some degree
of overlap between $K$ values for catchments consisting of different rock types (Fig. 6). However, a Kruskal-Wallis test shows that $K$ is statistically different in catchments underlain by different rock types at a 0.01 significance level. When the slope exponent $n$ is equal to 2, we find absolute values of $K$ to be more than one order of magnitude lower for every catchment. Nevertheless, the same relationship with rock type as the observed for the case of $n$ equal to 1 emerges, with low values of $K$ associated with quartzites and orders of magnitude higher $K$ values in areas in gneisses and granitic rocks (Fig. 6). Finally,
although the spatial variability in rainfall is limited in the study area, we observe that catchments that receive more precipitation



(in higher elevations underlain by resistant rocks) are associated with substantially lower $K$ values than areas that receive less precipitation (in more erodible rock units) (Fig. S5).

## 5 Discussion

### 5.1 Equilibrium as a natural attractor state in a decaying mountain belt

Our results show that bedrock strength controls the variability in topographic relief and channel steepness in the QF. This conclusion is consistent with the classic geomorphic expectation that, all else being equal, terrains underlain by resistant rocks will develop higher topographic relief and steeper channel gradients (e.g., Gilbert, 1877; Hack, 1960; Jansen et al., 2010; Bursztyn et al., 2015). Such correlation between topographic forms and rock strength is the same as that expected in an equilibrium adjustment scenario for a tectonically stable erosive landscape evolving over Myr timescales (Hack, 1960;

Montgomery, 2001). In the case of a spatially variable lithological configuration, theory predicts that more erodible rock units are progressively eroded whereas resistant rocks stand proud in relief, up to a point where differential topographic steepness everywhere balances spatial variations in rock strength and denudation rates are then spatially invariant (Hack, 1960). Our detrital cosmogenic-derived results, however, demonstrate that rates of denudation are kept spatially variable by the exposure at the surface of different rock types in a tectonically inactive landscape evolving for hundreds of Myr. The landscape has not

achieved any equilibrium or steady-state. We interpret our findings as an indication that quasi-equilibrium is likely not a natural attractor state in decaying mountain belts, or, at best, an attractor state that cannot be reached as long as lithological spatial heterogeneity (such as the one observed in the study area) is maintained, a conclusion that is consistent with the modelling results of Forte et al. (2016).

### 5.2 Lateral variations in rock type as a dominant control on denudation rates and topographic forms in a post-orogenic
**landscape**

Our results suggest that the fluvial erosion efficiency coefficient ($K$) varies as a function of rock type in the study area, with low $K$ values in resistant units and orders of magnitude higher $K$ values in more erodible rocks. In contrast, we find that spatial gradients in precipitation rates do not control changes in $K$ in the study area, considering that wetter climate conditions dictate higher values of $K$ (e.g., Ferrier et al., 2013). However, the fluvial erosion efficiency coefficient ($K$) as calculated here

incorporate other effects besides rock strength and precipitation rates, such as channel hydraulic geometry and sediment load (e.g., Snyder et al., 2000; Duvall et al., 2004; Zondervan et al., 2019). Whereas in the Appalachians effects on fluvial erosional efficiency such as channel width and sediment load has been shown to vary as a function of rock type (e.g., Spotila et al., 2015), we do not have data on such variables for the QF, which is a limitation to our reasonable interpretation that rock type controls $K$ in the study area. We note, however, that rock strength varies within each rock type in the study area and, for

instance, thin-bedded quartzites with large quantities of muscovite and sericite weather and erode much easier than average while some granitic areas stand bold in relief where these rocks are coarser-grained and more massive (Dorr, 1969).



The negative relationships we find between [10]Be-derived denudation rates with topographic relief, channel steepness and precipitation rates, appear to be contradictory to established theory, empirical studies and common sense (e.g., Ahnert, 1970; Montgomery and Brandon, 2002; Portenga and Bierman, 2011; Harel et al., 2016). Yet such relationships are consistent with the stream-power model if one accounts for the magnitude of variations in the fluvial erosion efficiency coefficient ($K$) estimated for the study area. Our findings are in agreement with studies that demonstrated that the link between denudation rates and channel steepness is obscure in settings where lateral variations in rock strength are important (e.g., Cyr et al., 2014; Campforts et al., 2020), and that a modified version of the stream-power model including variations in rock strength should be adopted for better predicting spatial patterns of channel incision (Campforts et al., 2020). Our results imply that modelling studies attempting to reconstruct uplift histories from river profile morphology assuming uniform fluvial erosion efficiency over large areas (e.g., Roberts and White, 2010) are likely to lead to flawed results, at least in post-orogenic settings.

To compare how our constraints on the fluvial erosion efficiency coefficient stand to published estimates of $K$, we also calculate $K$ in units of $m^{0.2}$/yr, as reported in several studies (e.g., Stock and Montgomery, 1999; Whipple et al., 2000b; Kirby and Whipple, 2001). We find that resulting estimates of $K$ in the QF (regional mean $K$ value of 7.1 x $10^{-7}$ $m^{0.2}$/yr, assuming the slope exponent $n = 1$) are comparable to the low-end or are lower than the $K$ values reported by Stock and Montgomery (1999) for a post-orogenic landscape underlain by granites and metasedimentary rocks in humid subtropical Australia (Fig. 7). Similarly, our estimates of $K$ (in Fig. 6) are comparable to the low-end or are lower than $K$ values estimated in crystalline and metamorphic rocks in the Upper Tennessee River (mean $K$ values of ~5 x $10^{-7}$ $m^{0.1}$/yr; Gallen, 2018), as well as in different physiographic provinces in the Appalachians such as the Blue Ridge (reported $K$ values: 7.8-3.0 x $10^{-7}$ $m^{0.1}$/yr; Gallen et al., 2013) and the Valley and Ridge (reported $K$ values: 2.1-1.5 x $10^{-6}$ $m^{0.1}$/yr; Miller et al., 2013). In contrast, our results are orders of magnitude lower than estimates of $K$ in tectonically active areas in Hawaii (tropical rainforest climate), California (Mediterranean climate), and Japan (humid continental climate; Stock and Montgomery, 1999), as well as the Siwalik Hills in Himalaya (monsoon highland climate; Kirby and Whipple, 2001), and the Ukak River in Alaska (cool continental climate; Whipple et al., 2000b).

It is difficult to isolate the influence of rock strength from climate conditions in this comparison. Nonetheless, Fig. 7 suggests a stark contrast over orders of magnitude in the fluvial erosion efficiency coefficient between tectonically active and inactive settings, which may result from the disparity between the high rates of denudation in orogenic belts, with exposure of mineral surfaces that are readily weathered, and the long timescales of evolution in ancient mountain belts, where the hard metamorphic roots of these landscapes are progressively exhumed as they get older (Summerfield and Hulton, 1994; Bishop, 2007; Braun et al., 2014; Bursztyn et al., 2015). In contrast, the global dataset of stream-power model parameters reported by Harel et al. (2016) exhibit lower $K$ values in tectonically active areas compared to higher $K$ values in inactive settings. These authors concluded that such results were counterintuitive and highlighted the fact that $K$ is not well calibrated (Harel et al., 2016); more work needs to be dedicated to the determination of $K$ and what controls its variations.



### 5.3 Landscape development in a decaying mountain belt marked by lateral contrasts in rock strength

Our findings indicate that high-relief uplands underlain by resistant bedrock are denuding more slowly than lower-relief surrounding areas associated with more erodible lithologies. Persistence of these denudation rates (averaged over timescales up to 1.1 Myr; Table S1) implies that relief in this ancient orogen is still growing rather than decaying. In this situation, relief grows not only because of the unequal denudation pattern across the landscape but also as a result of the flexural-isostatic compensation to the denudational unloading, which is a process that occurs at much wider wavelength than the local changes

in lithology and denudation rates (Gilchrist and Summerfield, 1990). Uplands and surrounding areas are equally isostatically uplifted in response to the regional denudation rate, resulting in a net reduction of mean elevation over time, but a slight increase in the heights of mountain peaks, similar to Molnar and England (1990). The timescale over which relief might continue to grow simply as a result of spatial variations in bedrock lithology is unresolved. However, our results suggest that relief is likely to persist long into the future in a landscape that would otherwise be suited to rapid denudation (high-relief, and

relatively high precipitation rates and warm climate).

The extrapolated effect of the denudation pattern in the QF, which is simplistic as it considers that geology does not vary with depth over the long timescales of post-orogenic evolution, does not suggest that topographic rejuvenation has not affected the study area, but instead that the exposition at the surface of rocks with significant contrasts in erodibility naturally favours the survival of relief in the absence of ongoing tectonics. On the contrary, we have compelling geomorphic evidence that at least

some decaying mountain belts underwent spatial and temporal changes in denudation rates as a response to different types of forcing (e.g., tectonic, lithologic, climatic, mantle-driven, and far-field driven) long after cessation of crustal thickening (e.g., Pazzaglia and Brandon, 1996; Quigley et al., 2007; Gallen et al., 2013; Tucker and van der Beek, 2013; Liu, 2014; Gallen, 2018). In this situation, large spatial contrasts in the fluvial erosion efficiency coefficient ($K$), such as the observed in our study area, determine significant differences in erosional response times across the landscape (i.e., the timescale of channel profile

response to perturbations in boundary conditions), with response times ($T_d$) decreasing as $K$ increases ($T_d$ scales to $1/K^n$, where $n$ is the slope exponent in the stream-power model; Baldwin et al., 2013), and thus spatial gradients in $K$ controls how post-orogenic landscapes respond to a giving force. Indeed, lithological controls on post-orogenic landscape dynamics have long been identified (e.g., Hack, 1960, 1975; Twidale, 1976; Mills, 2003; Bishop and Goldrick, 2010; Spotila et al., 2015; Gallen, 2018; Bernard et al., 2019; Vasconcelos et al., 2019) and, nonetheless, lithological contrasts have not been addressed

by numerical modelling of post-orogenic landscape evolution and, in particular, topographic decay (e.g., Baldwin et al., 2003; Egholm et al., 2013).

## 6 Conclusion

We present [10]Be concentrations in river sand from catchments spanning the range in topographic metrics and bedrock erodibility in a humid semitropical, tectonically inactive landscape in Brazil. The results of this study suggest that the post-

Earth **Surface**
Dynamics
Discussions

EGU

orogenic history of the study area is not a progressive reduction in relief and denudation rates. Instead, the exposition at the surface of rocks with strong lateral contrasts in erodibility amplifies spatial differences in topographic forms and denudation rates over time, which sustains or, indeed, increases relief in a tectonically dead landscape. We show that spatial variations in topographic relief and channel steepness can be explained by changes in rock type in the study area, and yet denudation rates are not uniformity distributed. Given the long period since the cessation of crustal thickening, we conjecture that the landscape

has not achieved equilibrium and that equilibrium is not a natural attractor in ancient landscapes. Our results indicate that the fluvial erosion efficiency differs by three orders of magnitude in the study area, varying as a function of rock type. This study demonstrates that lateral variations in rock strength play an essential role in the dynamics of an ancient mountain belt, and likely in other post-orogenic settings characterised by lithological spatial heterogeneity, in which they control the tempo and style of landscape response to changes in boundary conditions while also affecting their pattern of denudation.

**7 Data availability**

The data supporting the findings of this study are available in the Supplemental Material, including [10]Be analytical results and derived denudation rate data, catchment-averaged geomorphic parameters, and detailed information on catchment lithology. Extraction of topographic metrics and catchment-averaged pressure were carried out using LSDTopoTools version 2.03 (Mudd et al., 2020).

**8 Author contributions**

D.P. designed the study with contributions from all co-authors. D.P. and D.F. performed the cosmogenic isotope analysis. D.P. quantified the geomorphic parameters. D.P., C.P., M.D.H, P.B., and D.F. wrote the manuscript. D.P produced the Figures.

**9 Competing interests**

The authors declare no competing interests.

**10 Acknowledgements**

We thank the German Aerospace Center (DLR), the Natural Environment Research Council (NERC), and the Coordination for the Improvement of Higher Education Personnel (CAPES) for research support. We thank Hugh D. Sinclair for providing feedback on an early version of the manuscript.





## 11 Financial support

The DLR granted us access to TanDEM-X data as part of the project DEM_GEOL1345. NERC supported the cosmogenic isotope analysis under the CIAF award number 9177.0417. D.P. had support from CAPES under a Science without Borders fellowship (n° BEX 12000/13-2) and, subsequently, a CAPES-PrInt Postdoctoral fellowship (n° 88887.367976 / 2019-00).

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





## 13 Tables

**Table 1: Variability in $\theta$ in the QF.** Best-fit $\theta$ values were computed for all catchments of stream-order higher than third-order based on the disorder method (Hergarten et al., 2016) using code developed by Mudd et al. (2018).

| Stream-order | Number of basins | Best-fit $\theta$ statistics | |
| --- | --- | --- | --- |
| | | Mean $\theta$ | Standard deviation |
| fourth-order | 114 | 0.37 | 0.17 |
| fifth-order | 24 | 0.45 | 0.12 |
| sixth-order | 5 | 0.46 | 0.08 |
| seventh-order | 1 | 0.53 | - |
| **all catchments** | 144 | 0.44 | 0.13 |


Earth **Surface**
**Dynamics**
Discussions
**14 Figures**

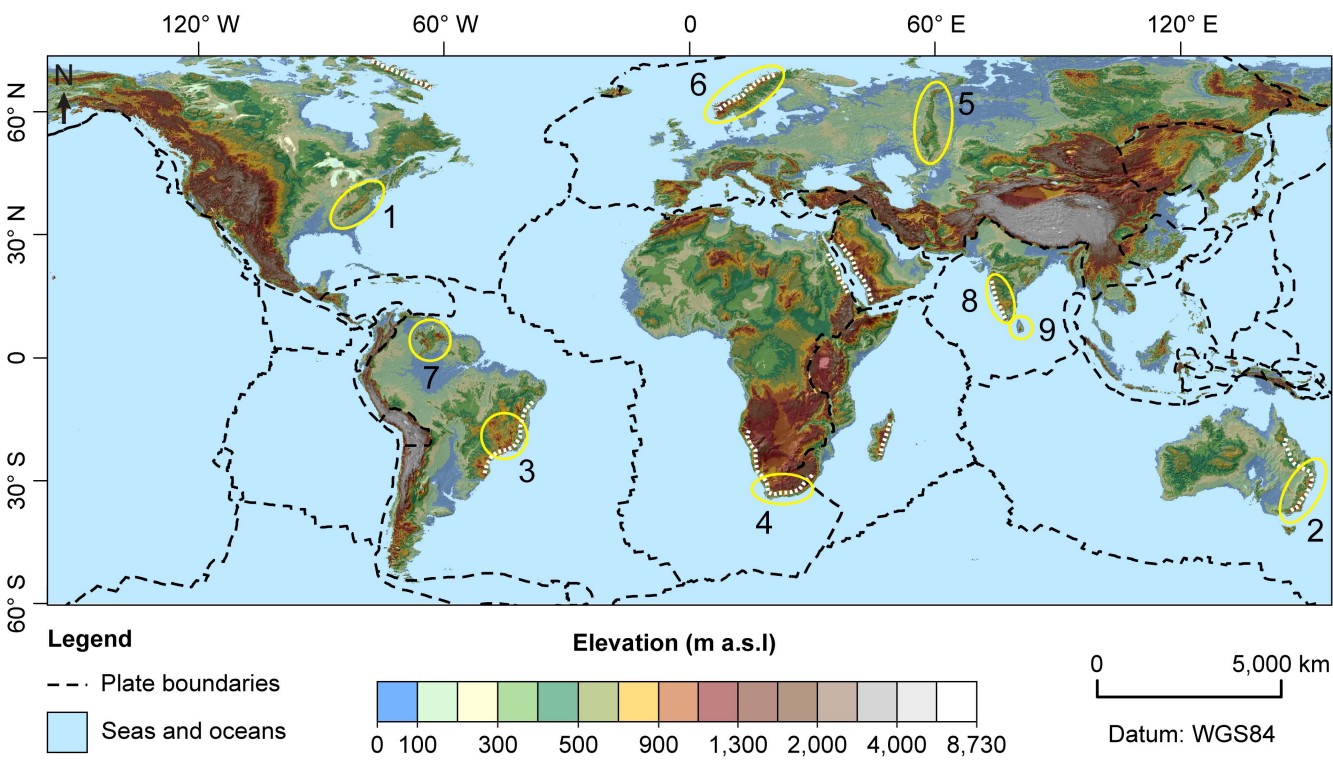

**Figure 1: Global elevation and striking examples of high-relief ancient mountain belts.** Post-orogenic settings highlighted by yellow ellipses: (1) the Appalachian Mountains; (2) SE Australia; (3) SE Brazil; (4) the Cape Mountains; (5) the Ural Mountains; (6) the Scandinavian Caledonides; (7) the Guyana Shield; (8) the Western Ghats; and (9) the Sri Lanka orogen. White dotted lines represent high-elevation passive margins. We extracted elevation data from the U.S. Geological Survey's (USGS) Global Multi-resolution Terrain Elevation
data 2010 (GMTED2010).





**Figure 2: The geomorphic context of the Quadrilátero Ferrífero – Brazil.** (A) Map of local topographic relief (extracted using a 2-km diameter window). (B) Simplified bedrock geology in the study area, comprising mostly steeply dipping (≥35°), Archean and Paleoproterozoic sequences. (C) Map of normalised channel steepness ($k_{sn}$) extracted using the segmentation method of Mudd et al., (2014), with $\theta_{ref}$ of 0.45. Note that we used an area threshold of 1.0 km² to extract the drainage network, and we did not show sampling sites in panel (C) for illustration purposes. (D) The location of the QF in Brazil, ~340 km away (in a straight line) from the Atlantic Ocean.



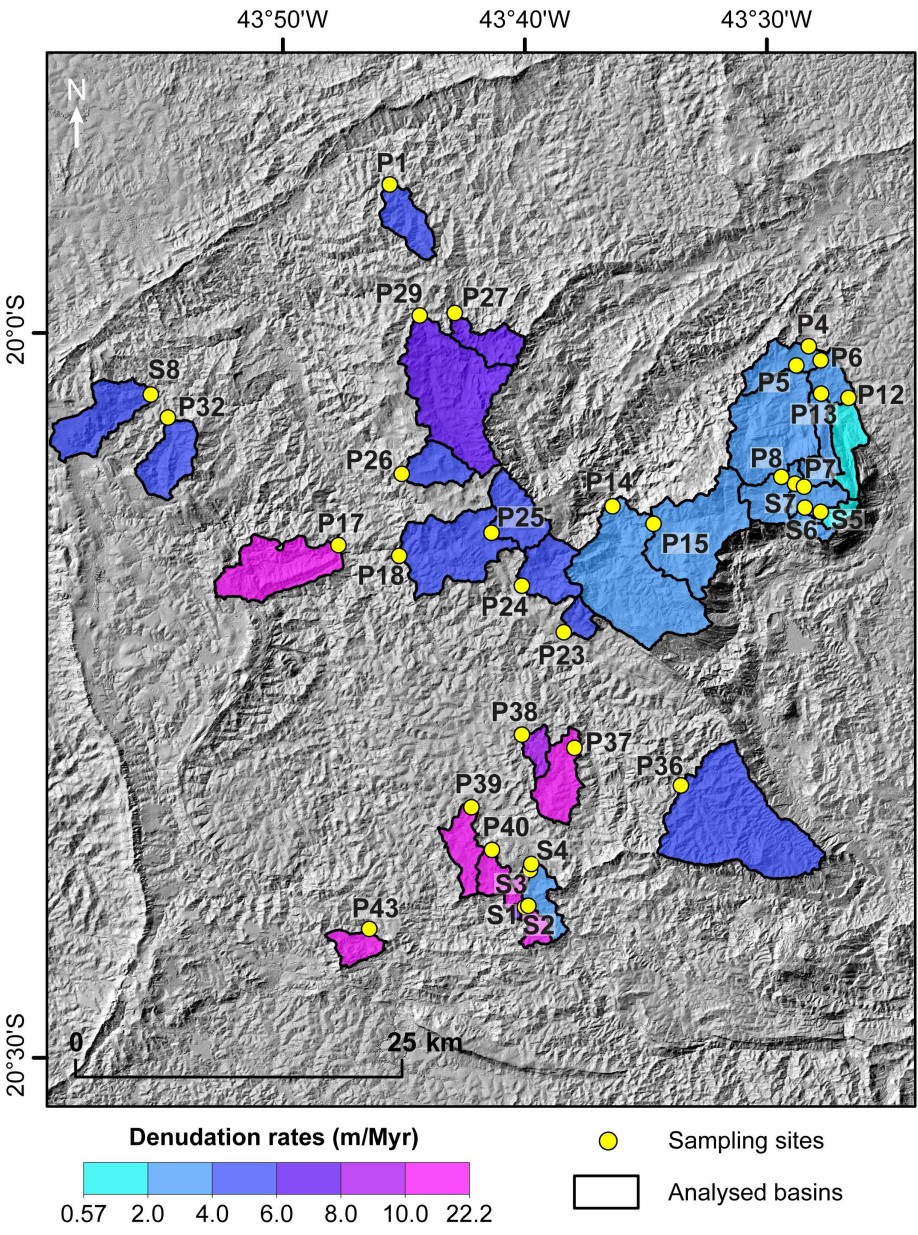

**Figure 3: Catchment-averaged denudation rates in the study area draped over a hillshade image.** Note that we identify as P'number' the 25 sampling sites of this study, whereas we label as S'number' the 8 sampling sites incorporated from Salgado et al. (2008). See Table S1 for $^{10}$Be analytical results and derived denudation rate data.





**Figure 4: Links between denudation rates, geomorphic parameters, and rock type in the study area.** Variations in catchment-averaged denudation rates with (A) mean local relief, (B) mean normalised channel steepness, (C) mean annual precipitation rates, and (D) percentage areal contribution of resistant rocks. Y-error bars show measurement errors in the nuclide concentration as well as errors related to the scaling method, and X-error bars indicate the SE of the mean. (E) Variations in catchment-averaged denudation rates per rock type, with the box on the left and raw data (diamonds) on the right. Box range represents the SE of the mean, whiskers show the interval between the 10th and 90th percentiles of the data, white squares show mean values, and thick black lines exhibit median values. Mixed lithology refers to catchments where a single lithology does not account for ≥75% of the catchment area.





**Figure 5: Rock type controls topographic forms in the study area.** Violin plots show the probability density (smoothed by a kernel density estimator) of (A) local relief, (B) normalised channel steepness, and (C) catchment-averaged local relief, per rock type. Panels (A) and (B) represent the distribution of local relief and normalised channel steepness for the entire study area, whereas panel (C) shows mean values of local relief for all catchments with stream-order equal or higher than second-order (Strahler, 1957) underlain by the rock types represented in panels (A)–(C). Whiskers show the interval between the 10th and 90th percentiles of the data, white squares represent mean values, and thick black lines exhibit median values.



Earth **Surface**
**Dynamics**
Discussions



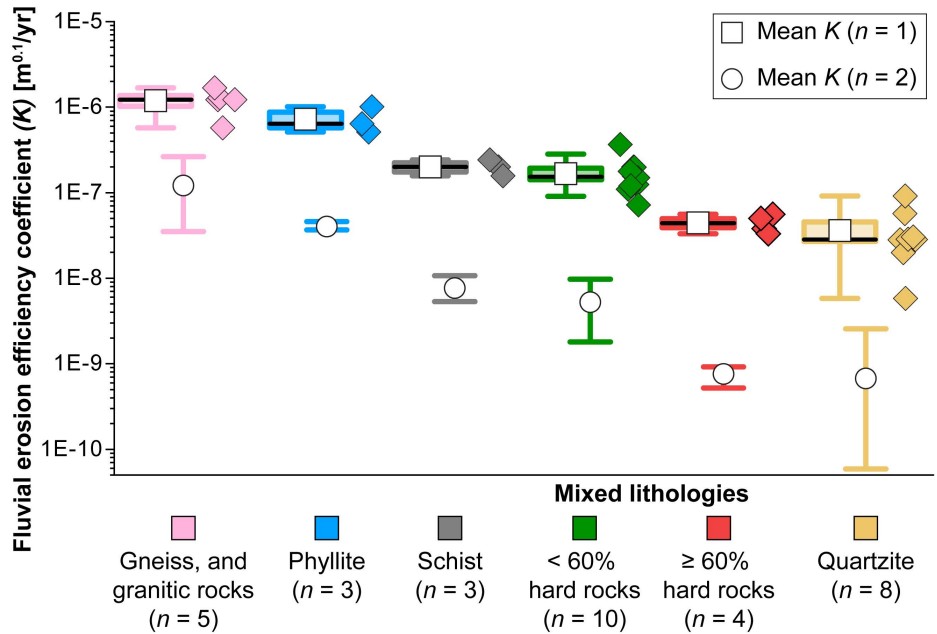

**Figure 6: Rock type controls variations in the fluvial erosion efficiency coefficient (*K*) in the study area.** Boxplot elements: box range represents the SE of the mean, whisker range shows the interval between the 10th and 90th percentiles of the data, and thick black line exhibit median values. Note that we calculated *K* assuming the slope exponent *n* = 1 and *n* = 2. For the case of *n* = 1, we show the boxplot on the left and raw data (diamonds) on the right. See Table S1 for data on each of the catchments analysed, including lithology and mean annual precipitation rates.

635





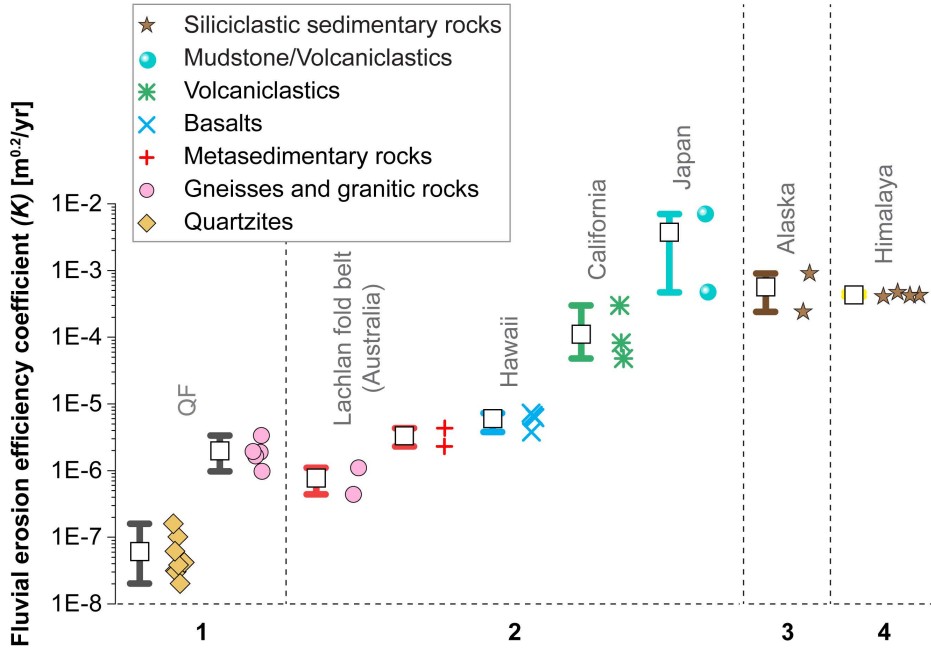

**Figure 7: Comparison of our results for the fluvial erosion efficiency coefficient (*K*) with previously published estimates of *K*.** Note that we display whiskers (representing the interval between the 10th and 90th percentiles of the data) and mean values (white squares) on the left and raw data (diamonds) on the right. Estimates of *K* were derived assuming the slope exponent *n* = 1. Numbers in the X-axis
640  represent different data sources: (1) this study; (2) Stock and Montgomery (1999); (3) Whipple et al., (2000b); (4) Kirby and Whipple (2001). Lithology was compiled as described by authors.