# Peer review of "Growing topography due to contrasting rock types in a tectonically dead landscape"

_Earth Surface Dynamics, 2020_

## Short Comment (SC1) · 26 Aug 2020

Hello, I just read through this and found it to be very interesting - there's not enough attention paid to tectonically dead landscapes. I recently published two papers that you might find to be relevant. I've attached them here. Cheers Manny

Please also note the supplement to this comment:
https://esurf.copernicus.org/preprints/esurf-2020-68/esurf-2020-68-SC1-supplement.zip

---

## Short Comment (SC2) · 18 Sep 2020

The preprint presents some great field, laboratory, and computational work, interprets the results in a reasonable way and provides insightful conclusions. While my current opinion of it is excellent, the first impression was not. It took me a lot of re-reading to figure out that it makes sense. Let me explain in order to help make the final article more attractive for casual readers as well.

The claim in line 240 "denudation rates are negatively correlated with normalised channel steepness" is surprising when one looks at Equation (5b) which implies that fluvial erosion and steepness are positively correlated. Is denudation negatively correlated with fluvial erosion? Is there a mistake? Steepness is not a quantity that can be measured in nature, it is derived and requires a choice of concavity. Could a poor choice lead to this unexpected result?

In fact, everything is fine. The context and Figure 4B tell that catchment-averaged normalised channel steepness is being discussed. Figures 4E (the link between denudation rate and rock type) and 5C (the link between the catchment-averaged local relief and rock type) provide results that come directly from measurements and are easier to interpret:

- hard rocks denude more slowly;

- as a consequence, relief on them is higher.

Channel steepness on hard rocks is thus higher as well (for a reasonable concavity). Therefore, there is a negative correlation between the steepness and the denudation rate when speaking of catchment averages.

Assuming I'm a typical reader, a typical reader would understand this point with less effort if the results from figures 4E and 5C (which directly describe nature and are in agreement with one's expectations) were mentioned first and emphasised more.

I believe the units for K depend on the exponent $m$ (see equation 4) and are not fixed for a given concavity. In this case, the claim in line 219 that the unit follows from the reference concavity is not exactly right. The results referred to around the line 272 with $n = 2$ may be given with a wrong unit, assuming the reference concavity was the same. Conversely, different concavity indices could result in the same unit for K, so not every K with the same unit has the same meaning (in contrast with what line 317 implies). All of it has no consequence for the conclusions of the article.

There are also a few little things I'd like to mention. The terms 'steepness' and 'steepness index' seem to mean the same thing, similarly for 'concavity' and 'concavity index'. Consistent use of one version would eliminate any doubt. The DOI of Perne et al., 2017

appears to be wrong. In the caption of the Figure S2, the description of subplots (A, C, E) should refer to (A, B, C). Regarding the lines 341 and 342, referring to persistence seems unnecessary for the relief to be growing (relief growth is not associated with a particular timescale so no particular averaging period is necessary). The persistence implies that relief has on average been growing throughout the averaging time scale.

---

## Referee Comment (RC1) · Anonymous Referee #1 · 21 Sep 2020

This is an innovative approach to a problem that has been around for a long time, and is worthy of publication. I have three substantive comments and a few minor ones. The first substantive comment is that if the denudation rate data were stratified according to rock type it might then be that relief will be a correlate with denudation rate. After all I assume that the authors are not suggesting that the physics of erosion no longer applies, including the sine of slope function. To make the claim that you have contradicted established theory on the basis of this partial analysis is not supportable. The second is that the term 'equilibrium' , 'steady state', and 'quasi-equilibrium' are used at many places without definition or explanation. This is a concern as, I am sure the authors know, the concept of equilibrium in geomorphology is, to say the least, vexed. What do you mean by these terms and how do you justify your usage? The third

substantive issue concerns denudation rate vs. averaging time. With one exception the denudation rates have averaging times less than 0.35Ma and there are a lot much less than 0.35Ma. It is necessary in my view to stratify the denudation rate data according to various averaging times to see if you get different results. You are asking a lot of an analysis that uses such a range of averaging times (27ka to 1.1Ma). And either exclude the rate at about 1.1Ma or explain it. I have added a graph of these data. My minor comments follow: 1.line 39. Is it still called the Lachlan Fold Belt? 2.line 76 What is semitropical? It is either tropical or it is not. 3.lines 96-101. I would like to see a little more information about the accuracy of these estimates and whether or not this is a craton. It is called an ancient orogen at line 143. 4. line 342 You cannot claim that the denudation rate has persisted fro 1.1Ma on the basis of the existing analysis (see substantive comment three above).The 1.1Ma value may be an anomaly. 5. line 343. Can you make this claim about flexural-isostatic compensation without modelling of this landscape? Or are you making an argument from theory. If the latter please make this clear.

**Brazil**

[Figure]

**Fig. 1.**

---

## Short Comment (SC3) · 6 Oct 2020

Dear Manny Gabet,

Thank you for such an encouraging comment.

It is fascinating that you are interested in the many intriguing questions that tectonically dead landscapes pose, and indeed they deserve more attention. I read your papers on the effects of lithological heterogeneity in the evolution of mountainous topography with enthusiasm; the reading was delightful, and it gave me many insights of future work.

Sincerely,

Daniel Peifer

[Figure]

**ESurfD**

Interactive
comment

---

## Author Comment (AC1) · 6 Oct 2020

Dear Matija Perne,

Thank you for your thoughtful and helpful comments; it is rewarding to receive your feedback that our manuscript is a valuable contribution. We are very grateful that you read our manuscript so carefully and took the time to write such constructive inputs. We will take your comments on board when revising the manuscript.

(i) Your suggestion "Assuming I'm a typical reader, a typical reader would understand this point with less effort if the results from figures 4E and 5C (which directly describe nature and are in agreement with one's expectations) were mentioned first and emphasised more" is excellent and very helpful; we will use it to frame our results better.

[Figure]

(ii) The critical point Matija is concerned in his comment of units for K is that we cannot directly compare K values for n=1 and n=2 for our fixed, topographically-informed concavity (theta =0.45), because the units change. This remark is correct, and we should be more cautious in how we report these results. We will revise the sentences addressing this comparison as well as the Y-axis label in Fig. 6. Nevertheless, the statement we make from the comparison is the most important, that changing n does not change the relationships we find between rock type, catchment-averaged denudation rates, and the fluvial erosion efficiency coefficient; we will revise accordingly to make it clear.

(iii) As suggested, we (a) will use "steepness" and "concavity" consistently in the revision; (b) we will make sure every DOI is correct; (c) we will fix the caption of the Figure S2, and (d) we agree with your remarks about 'persistence' and will revise its use.

Sincerely,

Daniel Peifer, on behalf of all co-authors

---

## Author Comment (AC2) · 8 Oct 2020

**Reviewer comment 1: *Interactive comment* on "Growing topography due to contrasting rock types in a tectonically dead landscape" by Daniel Peifer et al.**

**Daniel Peifer et al.**

peiferdaniel@gmail.com

**Response to review**

We thank the reviewer 1 for her/his thoughtful comments on our manuscript. Below are our responses, which will be changed in the final manuscript submission following completion of the interactive review period. We have numbered and italicised each reviewer comment, while our responses are coloured blue.

*1. This is an innovative approach to a problem that has been around for a long time, and is worthy of publication. I have three substantive comments and a few minor ones.*

We are pleased that the reviewer appreciates our work and perceives it as a valuable contribution. In the following, we address his/her comments and suggestions.

*2. The first substantive comment is that if the denudation rate data were stratified according to rock type it might then be that relief will be a correlate with denudation rate. After all I assume that the authors are not suggesting that the physics of erosion no longer applies, including the sine of slope function. To make the claim that you have contradicted established theory on the basis of this partial analysis is not supportable.*

All else being equal, steeper slopes should lead to more rapid denudation rates. Our study area, however, is characterised by considerable spatial variations in lithology, where resistant and more erodible rock types are exposed in a slowly eroding, humid

environment. Our results show that catchments underlain by what we infer to be resistant rocks, such as physically robust and chemically inert quartzites, are linked to higher catchment-averaged topographic metrics and lower catchment-averaged denudation rates than catchments in what we infer as more erodible rock types, such as gneisses and granitic rocks with abundant feldspars which are readily weathered in such climate conditions. In this situation, we do not claim to have contradicted established theory. Instead, we show that substantial lateral variations in (inferred) rock strength in a post-orogenic setting obscure any regional relationships between catchment-averaged denudation rates and basin-wide topographic metrics and precipitation rates that might otherwise exist. Our contribution highlights that lateral and vertical variations in rock strength (in our case, inferred) are essential players in post-orogenic landscape dynamics, which have been overlooked to some degree despite widespread assertions that lithological resistance is of fundamental importance in landscape evolution. And we welcome the fact that such a viewpoint is now receiving more attention in modelling and empirical studies (e.g., Forte et al., 2016; Perne et al., 2017; Gallen, 2018; Bernard et al., 2019; Strong et al., 2019; Vasconcellos et al., 2019; Campforts et al., 2020; Gabet, 2020a, 2020b; Zondervan et al., 2020a, 2020b). Nevertheless, we will remove our statement "appear to be contradictory to established theory" [line 308] in the revision.

Following the reviewer suggestion, we have included a figure showing variations in catchment-averaged denudation rates with mean normalised channel steepness (extracted using a reference concavity of 0.45) for individual rock types (Fig. R1). As expected, we observe that catchment-averaged denudation rates and mean normalised channel steepness may increase together for several rock types, though with such small

sample sizes no such relationships are statistically significant (at the $\alpha = 0.05$ level) except for catchments in phyllites. We conjecture that we did not find statistically significant positive relationships between these variables for every rock type because: (i) the relatively low range in values of topographic metrics for catchments underlain by the same rock types (for example, every catchment in gneisses and granite gneiss is characterised by low values of catchment-averaged normalised channel steepness); and (ii) internal variability in the fluvial erosion efficiency coefficient within each rock type (as discussed in the manuscript). Moreover, we note that the fluvial erosion efficiency coefficient incorporates controls other than rock type, which likely increases the internal variability in fluvial erosion efficiency in areas underlain by the same rock type. Nevertheless, we emphasise that we would expect denudation to increase together with topographic metrics in areas with the same fluvial erosion efficiency coefficient. We will add Figure R1 to the Supplemental Materials in the revision, referencing it in the Results section. Nonetheless, and we emphasise this fact, the relationships (for individual rock types) between catchment-averaged denudation rates and mean local relief, normalised channel steepness and annual precipitation rates are already apparent in Fig. 4.

[Figure]

**Figure R1: Variations in catchment-averaged denudation rates with mean normalised channel steepness for individual rock types.** Y-error bars show measurement uncertainties in the nuclide concentration as well as uncertainties related to the scaling method. Mixed lithology refers to catchments where a single lithology does not account for ≥75% of the catchment area.

***3.*** *The second is that the term 'equilibrium', 'steady state', and 'quasi-equilibrium' are used at many places without definition or explanation. This is a concern as, I am sure the authors know, the concept of equilibrium in geomorphology is, to say the least, vexed. What do you mean by these terms and how do you justify your usage?*

We agree with the reviewer that concepts such as "equilibrium" and "steady state" are best used when clearly defined. In this contribution, we refer to "equilibrium" and "steady state" implying a "topographic equilibrium" in which topographic forms are constant through time and denudation rates are spatially invariant irrespective of differences in rock type or topographic relief; in this situation, rock uplift is balanced by erosion, and topographic relief is adjusted to rock strength so that everywhere is downwasting at the same rate (Hack, 1960; Montgomery, 2001). The "topographic equilibrium" concept is somewhat problematic for post-orogenic landscapes given that rock uplift is necessary to maintain equilibrium (e.g., Kooi and Beaumont, 1996), yet some post-orogenic settings have been interpreted as in a topographic steady state, perhaps driven by isostatic denudational rebound (e.g. Matmon et al., 2003; Mandal et al., 2015). We described these concepts and interpretations in lines 65-69, which we can rephrase for clarity. As a full topographic steady state is likely never achieved (Willet and Brandon, 2002), when discussing our results we used the term "quasi-equilibrium" [line 290] meaning a less strict "topographic equilibrium" (where denudation rates should be nearly spatially invariant); we interpreted our results as an indication that our landscape has not achieved "topographic equilibrium" or "quasi-equilibrium".

*4.* *The third substantive issue concerns denudation rate vs. averaging time. With one exception the denudation rates have averaging times less than 0.35Ma and there are a lot much less than 0.35Ma. It is necessary in my view to stratify the denudation rate data according to various averaging times to see if you get different results. You are asking a lot of an analysis that uses such a range of averaging times (27ka to 1.1Ma). And either exclude the rate at about 1.1Ma or explain it. I have added a graph of these data.*

This remark is correct, and we did present one denudation rate estimate (sample ID: S5) with an average timescale much higher than the averaging timescale of all other denudation rate estimates. However, we do not consider such denudation rate estimate to be problematic for the conclusions of our study. First, measurements and averaging times (i.e., time taken for sand grains to be exhumed through the CRN production zone near the surface) are implicitly coupled and thus it is not possible to separate them; the slower the denudation rate, the longer the time averaged over. The "anomalous" denudation rate estimate (0.6 ± 0.1 m/Myr) was derived for a catchment in quartzite; all other estimates derived for catchments in quartzites yielded similarly low rates of denudation (ranging from 1.6 ± 0.2 to 3.3 ± 0.3 m/Myr). Thus there is no indication that such a denudation rate estimate is somehow incorrect. Second, when we remove the "anomalous" denudation rate estimate from our analysis, we find that all of the relationships previously found between catchment-averaged denudation rates and mean topographic metrics and precipitation rates hold (see the figure attached). That being said, our sentence "Persistence of these denudation rates (averaged over timescales up to 1.1 Myr; Table S1)" [line 341] is misleading, and we will remove such a statement in the revision.

[Figure]

**Figure R2: Links between denudation rates, geomorphic parameters, and rock type in the study area excluding sample S5.** Variations in catchment-averaged denudation rates with (A) mean local relief, (B) mean normalised channel steepness, (C) mean annual precipitation rates, and (D) percentage areal contribution of resistant rocks. Y-error bars show measurement uncertainties in the nuclide concentration as well as uncertainties related to the scaling method, and X-error bars indicate the SE of the mean. (E) Variations in catchment-averaged denudation rates per rock type, with the box on the left and raw data (diamonds) on the right. Box range represents the SE of the mean, whiskers show the interval between the 10th and 90th percentiles of the data, white squares show mean values, and thick black lines exhibit median values. Mixed lithology refers to catchments where a single lithology does not account for ≥75% of the catchment area.

**Minor comments**

*5. Line 39. Is it still called the Lachlan Fold Belt?*

It is still referred to as Lachlan Fold Belt to the best of our knowledge.

*6. Line 76. What is semitropical? It is either tropical or it is not.*

We agree with the reviewer. We will refer in the revision as "humid subtropical" given that climate over the study area ranges from Cwa to Cwb in Köppen-Geiger's classification (Alvares et al., 2013); we will add this information in the revision.

*7. Lines 96-101. I would like to see a little more information about the accuracy of these estimates and whether or not this is a craton. It is called an ancient orogen at line 143.*

Thank you for this remark. We will add in the revision information about the accuracy of geochronological data cited. The study area is a post-orogenic landscape with a polyphase deformation history, the last episode of which was ~500-450 Myr ago, and thus it is not a craton (see lines 91-93).

*8. Line 342 You cannot claim that the denudation rate has persisted from 1.1Ma on the basis of the existing analysis (see substantive comment three above). The 1.1Ma value may be an anomaly.*

Agreed. We will remove such reference in the revision (see response 4).

*9. Line 343. Can you make this claim about flexural-isostatic compensation without modelling of this landscape? Or are you making an argument from theory. If the latter please make this clear.*

We are making this argument from theory, and we will make it clear in the revision.

**References**

Alvares, C.A., Stape, J.L., Sentelhas, P.C., de Moraes Gonçalves, J.L. and Sparovek, G.: Köppen's climate classification map for Brazil, Meteorol. Z., 22, 711-728, https://doi.org/10.1127/0941-2948/2013/0507, 2013.

Bernard, T., Sinclair, H.D., Gailleton, B., Mudd, S.M. and Ford, M.: Lithological control on the post-orogenic topography and erosion history of the Pyrenees, Earth Planet. Sc. Lett., 518, 53-66, https://doi.org/10.1016/j.epsl.2019.04.034, 2019.

Campforts, B., Vanacker, V., Herman, F., Vanmaercke, M., Schwanghart, W., Tenorio, G.E., Willems, P. and Govers, G.: Parameterisation of river incision models requires accounting for environmental heterogeneity: insights from the tropical Andes, Earth Surf. Dynam., 8, 447-447, https://doi.org/10.5194/esurf-8-447-2020, 2020.

Cyr, A.J., Granger, D.E., Olivetti, V. and Molin, P.: Distinguishing between tectonic and lithologic controls on bedrock channel longitudinal profiles using cosmogenic $^{10}$Be erosion rates and channel steepness index, Geomorphology, 209, 27-38, https://doi.org/10.1016/j.geomorph.2013.12.010, 2014.

Forte, A.M., Yanites, B.J. and Whipple, K.X.: Complexities of landscape evolution during incision through layered stratigraphy with contrasts in rock strength, Earth Surf. Proc. Land., 41, 1736-1757, https://doi.org/10.1002/esp.3947, 2016.

Gabet, E.J.: Lithological and structural controls on river profiles and networks in the northern Sierra Nevada (California, USA), GSA Bulletin, 132, 655-667, https://doi.org/10.1130/B35128.1, 2020a.

Gabet, E.J.: River profile evolution by plucking in lithologically heterogeneous landscapes: Uniform uplift vs. tilting, Earth Surf. Proc. Land., 45, 1579–1588, 10.1002/esp.4832, 2020b.

Gallen, S.F.: Lithologic controls on landscape dynamics and aquatic species evolution in post-orogenic mountains, Earth Planet. Sc. Lett., 493, 150-160, https://doi.org/10.1016/j.epsl.2018.04.029, 2018.

Hack, J.T.: Interpretation of erosional topography in humid temperate regions, Am. J. Sci., 258, 80-97, 1960.

Kooi, H. and Beaumont, C.: Large-scale geomorphology: Classical concepts reconciled and integrated with contemporary ideas via a surface processes model. J. Geophys. Res.-Solid, 101, 3361-3386, https://doi.org/10.1029/95JB01861, 1996.

Mandal, S.K., Lupker, M., Burg, J.P., Valla, P.G., Haghipour, N. and Christl, M.: Spatial variability of 10Be-derived erosion rates across the southern Peninsular Indian escarpment: A key to landscape evolution across passive margins, Earth Planet. Sc. Lett., 425, 154-167, https://doi.org/10.1016/j.epsl.2015.05.050, 2015.

Matmon, A., Bierman, P.R., Larsen, J., Southworth, S., Pavich, M. and Caffee, M.: Temporally and spatially uniform rates of erosion in the southern Appalachian Great Smoky Mountains, Geology, 31, 155-158, https://doi.org/10.1130/0091-7613(2003)031<0155:TASURO>2.0.CO;2, 2003.

Montgomery, D.R.: Slope distributions, threshold hillslopes, and steady-state topography, Am. J. Sci., 301, 432-454, https://doi.org/10.2475/ajs.301.4-5.432, 2001.

Perne, M., Covington, M.D., Thaler, E.A. and Myre, J.M.: Steady state, erosional continuity, and the topography of landscapes developed in layered rocks, Earth Surf. Dynam., 5, 85-100, https://doi.org/10.5194/esurf-5-85-2017, 2017.

Strong, C.M., Attal, M., Mudd, S.M. and Sinclair, H.D.: Lithological control on the geomorphic evolution of the Shillong Plateau in Northeast India, Geomorphology, 330, 133-150, https://doi.org/10.1016/j.geomorph.2019.01.016, 2019.

Vasconcelos, P.M., Farley, K.A., Stone, J., Piacentini, T. and Fifield, L.K.: Stranded landscapes in the humid tropics: Earth's oldest land surfaces, Earth Planet. Sc. Lett., 519, 152-164, https://doi.org/10.1016/j.epsl.2019.04.014, 2019.

Willett, S.D. and Brandon, M.T.: On steady states in mountain belts, Geology, 30, 175-178, https://doi.org/10.1130/0091-7613(2002)030<0175:OSSIMB>2.0.CO;2, 2002

Zondervan, J.R., Stokes, M., Boulton, S.J., Telfer, M.W. and Mather, A.E.: Rock strength and structural controls on fluvial erodibility: Implications for drainage divide mobility in a collisional mountain belt. Earth Planet. Sc. Lett., 538, 1-13, https://doi.org/10.1016/j.epsl.2020.116221, 2020a.

Zondervan, J.R., Whittaker, A.C., Bell, R.E., Watkins, S.E., Brooke, S.A. and Hann, M.G: New constraints on bedrock erodibility and landscape response times upstream of an active fault, Geomorphology, 351, 1-14, https://doi.org/10.1016/j.geomorph.2019.106937, 2020b.

---

## Short Comment (SC4) · 29 Oct 2020

Two points:

Your study is exhumation of a passive margin landscape. There's some good work by Zondervan et al 2020 Earth and Planetary Science Letters who look at rock strength / erodibility in a collisional mountain belt (High Atlas, Morocco), but in a post orogenic phase (i.e. tectonically dead / quiescent). The relevance of this Zondervan work is that it is an inverted rift, with a post-rift, syn-rift cover and basement craton geology and thus the patterns and directions of exhumation are all rock strength and structure controlled.

The Brazilian passive margin is considered to be reactivated tectonically within the

[Figure]

Neogene, as per the very large volume of local Brazilian case study geomorphology / structural literature, which I think warrants some explanation.

---

## Editor Comment (EC1) · Greg Hancock (Editor) · 3 Dec 2020

Review of 'Growing topography due to contrasting rock types in a tectonically dead landscape' by Peifer et al An interesting and worthy topic of interest to the journal. The paper is of interest to both field workers and modellers. It pulls together, geology, climate and hydrology to unravel a well-understood concept and reverse the thinking. It is particularly interesting in that it presents an alternative view of denudation and relief. What is particular pleasing is an examination of the geology and geomorphology of a stable or dead landscape system. The paper is clearly written with an extensive list of references. The Abstract summarises the paper well. Some suggestions. In regards to the area-slope analysis and profile analysis, some extra detail and analysis may be extracted from Cohen et al (JGR, 2008 doi:10.1029/2007JF000820). It is not

clear what you mean by (line 376) 'Given the long period since the cessation of crustal thickening, we conjecture that the landscape has not achieved equilibrium and that equilibrium is not a natural attractor in ancient landscapes. Our results indicate that the fluvial erosion efficiency differs by three orders of magnitude in the study area, varying as a function of rock type.' and implications throughout the text. Is this not captured by the term declining equilibrium? The case for declining equilibrium has been argued by many at other sites globally. Minor issue: 'We' and 'our' in the Conclusion is repetitive.

---

## Author Response (AR1)

Daniel PEIFER

CAPES PrInt Postdoctoral Researcher

Institute of Geosciences (CPMTC - IGC - UFMG)

Federal University of Minas Gerais

peiferdaniel@gmail.com

Greg Hancock

Associate Editor, Earth Surface Dynamics

January 18, 2021

Dear Dr. Greg Hancock,

Thank you for considering our manuscript 'Growing topography due to contrasting rock types in a tectonically dead landscape' for publication in ESurf. We are very grateful to the editor, the reviewer, and our colleagues for providing constructive feedback, allowing us to improve the manuscript.

We have edited the manuscript and the supplemental materials to address all issues raised during the review process. Please find our response letters in the following, where original comments are numbered and italicised, and our responses are coloured blue. We also provide an edited version of the article where specific changes made to the manuscript are highlighted. Line numbers in our response letters refer to this version of the manuscript with tracked changes.

The main changes we have made to the manuscript are: (i) we emphasised that our results agree with our knowledge on erosion processes in terrestrial landscapes; (ii) we emphasised that we are comparing how *catchment-averaged* denudation rates vary with changes in *mean* values of topographic relief, channel steepness and precipitation rates; (iii) we excluded our reference to 'persistence' (of denudation rates) as an explanation for relief growth; (iv) we defined 'equilibrium' more clearly; (v) we made several small revisions to clarify points or correct typos; and (vi) we included two figures to the Supplemental Materials.

We hope these modifications have significantly improved the manuscript and we thank the editor, the reviewer and the scientific community for the thoughtful suggestions.

Best regards,

Daniel Peifer

**Table of Contents:**

**Response to RC1****:** *Interactive comment* **on "Growing topography due to contrasting rock types in a tectonically dead landscape" by Daniel Peifer et al.**

**Daniel Peifer et al.**

peiferdaniel@gmail.com

**Response to review**

We thank the reviewer 1 for her/his thoughtful comments on our manuscript.

*1. This is an innovative approach to a problem that has been around for a long time, and is worthy of publication. I have three substantive comments and a few minor ones.*

We are pleased that the reviewer appreciates our work and perceives it as a valuable contribution.

*2. The first substantive comment is that if the denudation rate data were stratified according to rock type it might then be that relief will be a correlate with denudation rate. After all I assume that the authors are not suggesting that the physics of erosion no longer applies, including the sine of slope function. To make the claim that you have contradicted established theory on the basis of this partial analysis is not supportable.*

All else being equal, steeper slopes should lead to more rapid denudation rates. Our study area, however, is characterised by considerable spatial variations in lithology, where resistant and more erodible rock types are exposed in a slowly eroding, humid environment. Our results show that catchments underlain by what we infer to be resistant rocks, such as physically robust and chemically inert quartzites, are linked to higher catchment-averaged topographic metrics and lower catchment-averaged denudation rates than catchments in what we infer as more erodible rock types, such as gneisses and granitic rocks with abundant feldspars which are readily

weathered in such climate conditions. In this situation, we do not claim to have contradicted established theory. Instead, we show that substantial lateral variations in (inferred) rock strength in a post-orogenic setting obscure any regional relationships between catchment-averaged denudation rates and basin-wide topographic metrics and precipitation rates that might otherwise exist. Our contribution highlights that lateral and vertical variations in rock strength (in our case, inferred) are essential players in post-orogenic landscape dynamics, which have been overlooked to some degree despite widespread assertions that lithological resistance is of fundamental importance in landscape evolution. And we welcome the fact that such a viewpoint is now receiving more attention in modelling and empirical studies (e.g., Forte et al., 2016; Perne et al., 2017; Gallen, 2018; Bernard et al., 2019; Strong et al., 2019; Vasconcellos et al., 2019; Campforts et al., 2020; Gabet, 2020a, 2020b; Zondervan et al., 2020a, 2020b). Nevertheless, we removed our statement "appear to be contradictory to established theory, empirical studies and common sense", modifying the sentence to [lines 331-335]: "*The negative relationships we find between [10]Be-derived catchment-averaged denudation rates and catchment-averaged values of topographic relief, channel steepness and precipitation rates, are counter-intuitive. However, such relationships are consistent with the stream-power model if one accounts for the magnitude of variations in the fluvial erosion efficiency coefficient (K) estimated for the study area.*"

Following the reviewer suggestion, we have included a figure showing variations in catchment-averaged denudation rates with mean normalised channel steepness (extracted using a reference concavity of 0.45) for individual rock types (Fig. AR1). As expected, we observe that catchment-averaged denudation rates and mean normalised channel steepness may increase together for several rock types, though with such small sample sizes no such relationships are

statistically significant at the $\alpha = 0.05$ level except for catchments in phyllites. We conjecture that we did not find statistically significant positive relationships between these variables for every rock type because: (i) the relatively low range in values of topographic metrics for catchments underlain by the same rock types (for example, every catchment in gneisses and granite gneiss is characterised by low values of catchment-averaged normalised channel steepness); and (ii) internal variability in the fluvial erosion efficiency coefficient within each rock type (as discussed in the manuscript). Moreover, we note that the fluvial erosion efficiency coefficient incorporates controls other than rock type, which likely increases the internal variability in fluvial erosion efficiency in areas underlain by the same rock type. Nevertheless, we emphasise that we would expect denudation to increase together with topographic metrics in areas with the same fluvial erosion efficiency coefficient.

We have added Figure AR1 to the Supplemental Materials (as Fig. S5), referencing it in the Results section through an additional sentence [lines 264-266]: "*However, we observe that catchment-averaged denudation rates may increase together with mean values of topographic metrics and precipitation rates for individual rock types, although with such small sample sizes no such relationships are statistically significant at the $\alpha = 0.05$ level except for catchments in phyllites (Fig. 4, S6).*"

[Figure]

**Figure AR1: Variations in catchment-averaged denudation rates with mean normalised channel steepness for individual rock types.** Y-error bars show measurement uncertainties in the nuclide concentration as well as uncertainties related to the scaling method. Mixed lithology refers to catchments where a single lithology does not account for ≥75% of the catchment area.

**3.** The second is that the term' equilibrium',' steady state', and' quasi-equilibrium' are used at many places without definition or explanation. This is a concern as, I am sure the authors know, the concept of equilibrium in geomorphology is, to say the least, vexed. What do you mean by these terms and how do you justify your usage?

We agree with the reviewer that concepts such as 'equilibrium' and 'steady state' are best used when clearly defined. In this contribution, we refer to 'equilibrium' and 'steady state' implying a 'topographic equilibrium' in which topographic forms are constant through time and denudation rates are spatially invariant irrespective of differences in rock type or topographic relief; in this situation, rock uplift is balanced by erosion, and topographic relief is adjusted to rock strength so that everywhere is downwasting at the same rate (Hack, 1960; Montgomery, 2001). The 'topographic equilibrium' concept is somewhat problematic for post-orogenic landscapes given that rock uplift is necessary to maintain equilibrium (e.g., Kooi and Beaumont, 1996), yet some post-orogenic settings have been interpreted as in a topographic steady state, perhaps driven by isostatic denudational rebound (e.g. Matmon et al., 2003; Mandal et al., 2015). We rephrased our description of these concepts and interpretations to [lines 68-72]: "*These observations were interpreted, in several cases, as equilibrium adjustments where spatial variations in rock strength are balanced by variations in topographic relief so that everywhere is eroding at the same rate, with the corollary that topographic forms are constant through time in a 'topographic equilibrium' likely driven by isostatic uplift (e.g., Hack, 1960; Matmon et al., 2003; Scharf et al., 2013; Mandal et al., 2015).*" To ensure consistency in using these concepts, we have excluded the term 'quasi-equilibrium' from the manuscript [line 313].

**4.** The third substantive issue concerns denudation rate vs. averaging time. With one exception the denudation rates have averaging times less than 0.35Ma and there are a lot much less than

0.35Ma. It is necessary in my view to stratify the denudation rate data according to various averaging times to see if you get different results. You are asking a lot of an analysis that uses such a range of averaging times (27ka to 1.1Ma). And either exclude the rate at about 1.1Ma or explain it. I have added a graph of these data.

This remark is correct, and we did present one denudation rate estimate (sample ID: S5) with an average timescale much higher than the averaging timescale of all other denudation rate estimates. However, we do not consider such denudation rate estimate to be problematic for the conclusions of our study. First, measurements and averaging times (i.e., time taken for sand grains to be exhumed through the CRN production zone near the surface) are implicitly coupled and thus it is not possible to separate them; the slower the denudation rate, the longer the time averaged over. The "anomalous" denudation rate estimate (0.6 ± 0.1 m/Myr) was derived for a catchment in quartzite; all other estimates derived for catchments in quartzites yielded similarly low rates of denudation (ranging from 1.6 ± 0.2 to 3.3 ± 0.3 m/Myr). Thus, there is no indication that such a denudation rate estimate is somehow incorrect. Second, when we remove the "anomalous" denudation rate estimate from our analysis, we find that all of the relationships previously found between catchment-averaged denudation rates and mean topographic metrics and precipitation rates hold (see Fig. AR2). That being said, our sentence "Persistence of these denudation rates (averaged over timescales up to 1.1 Myr; Table S1)" [line 367-368] is misleading, and we removed it from the manuscript. We modified the manuscript to [lines 365-368]: "*Our findings indicate that high-relief uplands underlain by resistant bedrock are denuding more slowly than lower-relief surrounding areas associated with more erodible lithologies, with the corollary that topographic relief must still be growing instead of decaying in this tectonically quiescent landscape*".

[Figure]

**Figure AR2: Links between denudation rates, geomorphic parameters, and rock type in the study area excluding sample S5.** Variations in catchment-averaged denudation rates with (A) mean local relief, (B) mean normalised channel steepness, (C) mean annual precipitation rates, and (D) percentage areal contribution of resistant rocks. Y-error bars show measurement uncertainties in the nuclide concentration as well as uncertainties related to the scaling method, and X-error bars indicate the SE of the mean. (E) Variations in catchment-averaged denudation rates per rock type, with the box on the left and raw data (diamonds) on the right. Box range represents the SE of the mean, whiskers show the interval between the 10th and 90th percentiles of the data, white squares show mean values, and thick black lines exhibit median values. Mixed lithology refers to catchments where a single lithology does not account for ≥75% of the catchment area.

**Minor comments**

*5. Line 39. Is it still called the Lachlan Fold Belt?*

It is still referred to as Lachlan Fold Belt to the best of our knowledge. However, we modified the

manuscript to [lines 38-40]: "*Examples of ancient mountain belts marked by high elevations and*

*steep slopes include the Appalachian Mountains, several mountain ranges SE Brazil, the Cape Mountains, parts of the East Australian Highlands, the Ural Mountains, the Caledonides, the Western Ghats, and the Sri Lanka orogen (Fig. 1).*"

**6.** *Line 76. What is semitropical? It is either tropical or it is not.*

We agree with the reviewer. We refer in the revised manuscript as "humid subtropical" [lines 14, 80, 119, 198, 272, 397] given that climate over the study area ranges from Cwa to Cwb in Köppen-Geiger's classification (Alvares et al., 2013). We modified the manuscript to include this information [lines 94-95], and we included Alvares et al. (2013) in our references.

**7.** *Lines 96-101. I would like to see a little more information about the accuracy of these estimates and whether or not this is a craton. It is called an ancient orogen at line 143.*

Thank you for this remark.

i) We modified the manuscript to describe the available data more thoroughly as [lines 102-111]: "*An array of geochronological data imply that the current topography is long-lived. These data include: a relatively large set of (U-Th)/He data (n = 291) and cosmogenic $^3$He concentrations (n = 71) in iron oxides showing mineral precipitation ages as old as 55 Ma (varying from 55.3 ± 5.5 to 0.4 ± 0.1 Ma), and exposure ages ranging from 10.9 ± 1.2 to 0.2 ± 0.1 Ma, with age versus elevation relationships suggesting older ages and a more extended history of exposure in iron duricrust-covered plateaus at higher elevations (Monteiro et al., 2014, 2018); $^{40}$Ar/$^{39}$Ar dating of Mn oxide grains collected in nine weathering profiles in the QF (n = 174 grains that produced reliable results) yielding ages ranging from 94.6 ± 5.5 to 12.3 ± 0.5 Ma, yet predominantly distributed between 30-60 Ma (Spier et al., 2006; Vasconcelos and Carmo, 2018); and low denudation rates (<3 m/Myr) implied by cosmogenic $^3$He and $^{10}$Be inventories (Monteiro et al.,*

*2018; Salgado et al., 2008).*" We have added Monteiro et al. (2014) and Spier et al. (2006) to our references.

ii) The study area is a post-orogenic landscape with a polyphase deformation history, the last episode of which was ~500-450 Myr ago, and thus it is not a craton (see lines 97-99). We have not made any modifications to the manuscript based on this point as we feel that it is already set out clearly in the manuscript.

**8.** *Line 342 You cannot claim that the denudation rate has persisted from 1.1Ma on the basis of the existing analysis (see substantive comment three above). The 1.1Ma value may be an anomaly.*

Agreed. We removed such reference from the manuscript [lines 367-370] (see response 4).

**9.** *Line 343. Can you make this claim about flexural-isostatic compensation without modelling of this landscape? Or are you making an argument from theory. If the latter please make this clear.*

We are making this argument from theory. We modified the manuscript to [lines 365-374]: "*Our findings indicate that high-relief uplands underlain by resistant bedrock are denuding more slowly than lower-relief surrounding areas associated with more erodible lithologies, with the corollary that topographic relief must still be growing instead of decaying in this tectonically quiescent landscape. Furthermore, the expected isostatic compensation to denudational unloading (e.g., Bishop and Brown, 1992), which is a process that occurs at a much longer wavelength than the local changes in lithology and denudation rates (Gilchrist and Summerfield, 1990; Watts et al. 2000), implies that uplands and surrounding areas are equally isostatically uplifted in response to the regional denudation, likely resulting in a net reduction of mean*

*elevation over time, but a slight increase in the heights of mountain peaks, as has been proposed by Molnar and England (1990).*" We have also added Watts et al. (2000) to our references.

  • *hard rocks denude more slowly;*

  • *as a consequence, relief on them is higher.*

*Channel steepness on hard rocks is thus higher as well (for a reasonable concavity). Therefore, there is a negative correlation between the steepness and the denudation rate when speaking*

*of catchment averages. Assuming I'm a typical reader, a typical reader would understand this point with less effort if the results from figures 4E and 5C (which directly describe nature and are in agreement with one's expectations) were mentioned first and emphasised more.*

Dear Matija Perne,

Thank you for such thoughtful and helpful comments; it is rewarding to receive your feedback that our manuscript is a valuable contribution. We are very grateful that you read our manuscript so carefully and took the time to write such constructive inputs.

We took your comments on board when revising the manuscript. First, we modified the manuscript in several places [lines 15, 255, 258, 261, and 263] to emphasise that we are comparing how catchment-averaged denudation rates vary with changes in mean values of local relief, normalised channel steepness and mean annual precipitation.

Second, we have added a sentence at the end of the first paragraph of the 'results' section to emphasise that our results indeed agree with our knowledge on erosion processes in terrestrial landscapes [lines 264-266]: "*However, we observe that catchment-averaged denudation rates may increase together with mean values of topographic metrics and precipitation rates for individual rock types, although with such small sample sizes no such relationships are statistically significant at the α = 0.05 level except for catchments in phyllites (Fig. 4, S6).*" We have also added Figure AR1 to the Supplemental Materials (see response 2 to RC1).

*3. I believe the units for K depend on the exponent m (see equation 4) and are not fixed for a given concavity. In this case, the claim in line 219 that the unit follows from the reference concavity is not exactly right. The results referred to around the line 272 with n = 2 may be given with a wrong unit, assuming the reference concavity was the same. Conversely, different*

*concavity indices could result in the same unit for K, so not every K with the same unit has the same meaning (in contrast with what line 317 implies). All of it has no consequence for the conclusions of the article.*

The critical point Matija is concerned in his comment of units for $K$ is that we cannot directly compare $K$ values for $n = 1$ and $n = 2$ for our fixed, topographically-informed concavity (concavity = 0.45), because the units change. This remark is correct, and we should be more cautious in how we report these results.

We modified the sentences addressing this comparison to: i) [lines 234-235] "*We extracted $k_{sn}$ using 0.45 as the reference concavity which, assuming m = 0.45 and n = 1, yields K values with units of $m^{0.1}$/yr.*"; and ii) [lines 293-295] "*When the slope exponent n is equal to 2, we find absolute values of K to be more than one order of magnitude lower for every catchment (assuming m = 0.9 and n = 2, derived values of K have units of $m^{-0.8}$/yr).*" We have also modified Fig. 6, so it shows the correct units. Nevertheless, the statement we make from the comparison is the most important, that changing $n$ does not change the relationships we find between catchment lithology, catchment-averaged denudation rates, and the fluvial erosion efficiency coefficient.

Although we agree with Matija's statement that "different concavity indices could result in the same unit for $K$, so not every $K$ with the same unit has the same meaning", for a given $m$ (for instance, 0.4), not every choice of $n$ result in reasonable values of concavity. For example, assuming $m = 0.4$ (and the units for $K$ depend on this exponent $m$), for $n = 1$ (concavity = 0.4, which is a reasonable value); whereas for $n = 2$ (concavity = 0.2); $n = 3$ (concavity = 0.1333333333333333); and $n = 4$ (concavity = 0.1). Thus, we have not changed our statement [lines 342-344]: "*To compare our constraints on the fluvial erosion efficiency coefficient with*

*published estimates of K, we also calculate K in units of $m^{0.2}/yr$, as reported in several studies*

*(e.g., Stock and Montgomery, 1999; Whipple et al., 2000b; Kirby and Whipple, 2001)."*

*4. There are also a few little things I'd like to mention. The terms' steepness' and' steepness index' seem to mean the same thing, similarly for' concavity' and' concavity index'. Consistent use of one version would eliminate any doubt.*

Agreed. We have now a consistent use of 'channel steepness' and 'channel concavity' in the manuscript.

*5. The DOI of Perne et al., 2017 appears to be wrong.*

Thanks for this remark. We made sure every DOI is correct in the revised manuscript.

*6. In the caption of the Figure S2, the description of subplots (A, C, E) should refer to (A, B, C).*

Fixed.

*7. Regarding the lines 341 and 342, referring to persistence seems unnecessary for the relief to be growing (relief growth is not associated with a particular timescale so no particular averaging period is necessary). The persistence implies that relief has on average been growing throughout the averaging time scale.*

Agreed. We excluded our reference to 'persistence', modifying our sentence to [lines 365-368]: "*Our findings indicate that high-relief uplands underlain by resistant bedrock are denuding more slowly than lower-relief surrounding areas associated with more erodible lithologies, with the corollary that topographic relief must still be growing instead of decaying in this tectonically quiescent landscape*".

**Response to SC4:** *Interactive comment* **on "Growing topography due to contrasting rock types in a tectonically dead landscape" by Daniel Peifer et al.**

**Daniel Peifer et al.**

peiferdaniel@gmail.com

**Response to SC4**

*1. Two points:*

*Your study is exhumation of a passive margin landscape. There's some good work by Zondervan et al 2020 Earth and Planetary Science Letters who look at rock strength / erodibility in a collisional mountain belt (High Atlas, Morocco), but in a post orogenic phase (i.e. tectonically dead / quiescent). The relevance of this Zondervan work is that it is an inverted rift, with a post-rift, syn-rift cover and basement craton geology and thus the patterns and directions of exhumation are all rock strength and structure controlled.*

Dear Martin Stokes,

Thank you very much for your constructive comments.

We agree that Zondervan et al. (2020) is relevant to our manuscript, particularly as it demonstrates (using robust methods that are important to our manuscript) that variations in rock strength (and its influence in the fluvial erodibility) play a fundamental role in the dynamics and evolution of a collisional mountain belt in a post-orogenic phase. Therefore, we have added Zondervan et al. (2020) to our references.

*2. The Brazilian passive margin is considered to be reactivated tectonically within the Neogene, as per the very large volume of local Brazilian case study geomorphology / structural literature, which I think warrants some explanation.*

Our study area, the Quadrilátero Ferrífero (QF), is not a passive margin. The QF lies in the southeastern edge of the São Francisco Craton (SFC; Fig. AR3), far away from the coastline (~350 km in a straight line from the Atlantic Ocean; see Fig. 2), in the boundary between the SFC and the Mantiqueira Province. The SFC consists of an Archean-Paleoproterozoic block that has not experienced major tectonic and magmatic events since ~1900 Ma (Almeida et al., 1981; Alkmim and Martins-Neto, 2012; Aguilar et al., 2017) encircled (on all sides) by Neoproterozoic-to Early Ordovician Brasiliano orogenic belts that developed during the Brasiliano/Pan-African collage of West Gondwana (Endo and Fonseca, 1992; Alkmim and Teixeira, 2017; Heilbron et al., 2017).

[Figure]

**Figure AR3: Simplified geology of the QF.** (A) The tectonic context of the assembly of West Gondwana by the end of the Proterozoic: cratons (in purple) surrounded by Neoproterozoic mobile belts (in yellow); in this reconstruction, the São Francisco Craton includes the Congo Craton, now in Africa. Note that there is some controversy about the position of the SFC's limit. Cratons: A – Amazonian; WA – West Africa; SFC – São Francisco-Congo; K – Kalahari. Modified from Alkmim and Martins-Neto (2012).

The QF experienced a polyphase deformation history resulting in a complex structural framework which is generally considered to be the result of three different kinematic phases identified using cross-cutting relationships and kinematic criteria (e.g., Chemale Jr et al., 1994;

Alkmim and Marshak, 1998) yet with limited absolute age constraints (Farina et al., 2016; Alkmim and Teixeira, 2017). In summary, the oldest phase is a Rhyacian collision related to a set of NE-SW-trending, NW-verging regional-scale folds (Alkmim and Marshak, 1998; Farina et al., 2016; Alkmim and Teixeira, 2017). The second phase is related to an extensional event associated with the formation of a dome-and-keel geological architecture, comprising basement domes and surrounding supracrustal synclines, likely in a context of the extensional collapse of the Rhyacian orogen (Chemale Jr et al., 1994; Alkmim and Marshak, 1998). The youngest phase is a Neoproterozoic- to Early Ordovician Brasiliano (650-480 Ma) compressional event, related to the amalgamation of West Gondwana, with the development of a west-verging thrust system that overprinted and reactivated pre-existing structures (Chemale Jr et al., 1994; Alkmim and Marshak, 1998; Alkmim and Teixeira, 2017). This compressional event affected mainly the eastern part of the QF, with the intensity of deformation decreasing westward; a series of WNW-verging faults and thrust cuts the entire Precambrian section of the eastern part of the QF (Chemale Jr et al., 1994; Alkmim and Marshak, 1998; Alkmim and Teixeira, 2017).

There is, indeed, a robust body of observational constraints, including thermochronological, sedimentary and geomorphic data, indicating a complex post-rift tectonic reactivation scenario in the elevated passive margin of SE Brazil, yet the available data suggest a much more stable tectonic history in the continental interior (e.g., Gallagher et al., 1994; Carmo, 2005; Tello Saenz et al., 2005; Hiruma et al., 2010; Cogné et al., 2011, 2012; Jelinek et al., 2014; Engelman de Oliveira et al., 2016; Krob et al., 2019; van Ranst et al., 2020; Fonseca et al., 2020). In this situation, the QF is not part of the elevated passive margin of SE Brazil, being located, instead, more in the deep interior of the continent (see Fig. 2), and there is no evidence that the QF was

directly affected by the continental breakup. Furthermore, the geochronological data available indicate that the study area's topography is long-lived (see lines 102-111).

Nonetheless, some authors did hypothesise that some areas in the QF, mainly the eastern half of the study area, experienced Cenozoic deformation based on field evidence of post-deformation in small and spatially restricted Cenozoic deposits (e.g., Saadi, 1992; Sant'anna et al.,1997; see lines 111-113); see the attached Fig. AR4 for the (limited) spatial distribution of Cenozoic units in the QF. Our sampling design included catchments spanning the range of topographic relief and bedrock lithologies in the study area, yet much of our data is derived from catchments located in the easternmost part of the QF (i.e., catchments P4, P5, P6, P7, P8, P12, P13, S5, S6 and S7; see Fig. 3). These catchments cover the area with the most pronounced topographic relief in the QF (Fig. 2), which is also characterised by the presence of many old structural anisotropies resulting from intense deformation experienced by this region during the Neoproterozoic- to Early Ordovician Brasiliano compressional event (Chemale Jr et al., 1994; Alkmim and Marshak, 1998; Alkmim and Teixeira, 2017). Our results show that these catchments in quartzites (i.e., catchments P7, P8, P12, P13, S5, S6 and S7) denude at low rates irrespective of their high topographic relief, with a minimum denudation rate of 0.6 ± 0.1 m/Myr for the catchment with the most pronounced topography (see Table S1). We did not observe any evidence of the influence of ongoing tectonic activity in our [10]Be-derived catchment-averaged denudation rates.

We modified the first sentence of the 'geologic setting' section to [lines 86-87]: "*The study area is the Quadrilátero Ferrífero (Brazil), one of Brazil's highest elevation areas, with a peak elevation of 2,076 m, located in the continental interior ~350 km away in a straight line from the Atlantic Ocean (Fig. 2).*" We have also added Fig. AR4 to the Supplemental Materials (as Fig.

S1) [see lines 111-114]. However, we have not made any other modifications to the manuscript based on these statements as we feel that the context and implications of our results are already set out clearly in the manuscript.

[Figure]

**Figure AR4: Spatial distribution of Cenozoic units in the QF, excluding iron duricrusts**. Geological data: Lobato et al., 2005. The Fonseca formation refers to Eocene clayey and sandy lacustrine deposits often interpreted as tectonically controlled (e.g., Sant' anna et al., 1997).

Dear Manny Gabet,

Thank you for such an encouraging comment.

It is fascinating that you are interested in the many intriguing questions that tectonically dead landscapes pose, and indeed they deserve more attention. I read your papers on the effects of lithological heterogeneity in the evolution of mountainous topography with enthusiasm; the reading was delightful, and it gave me many insights of future work. Again, thank you very much.

Sincerely,

Daniel Peifer

[revised manuscript text omitted]